DOI: 10.1038/s41467-018-05724-1　　**OPEN**

# Gating mechanism of the extracellular entry to the lipid pathway in a TMEM16 scramblase

Byoung-Cheol Lee[1,5], George Khelashvili[2,3], Maria Falzone[4], Anant K. Menon [4],
Harel Weinstein[2,3] & Alessio Accardi [1,2,4]

Members of the TMEM16/ANO family of membrane proteins are $Ca^{2+}$-activated phospholipid scramblases and/or $Cl^-$ channels. A membrane-exposed hydrophilic groove in these proteins serves as a shared translocation pathway for ions and lipids. However, the mechanism by which lipids gain access to and permeate through the groove remains poorly understood. Here, we combine quantitative scrambling assays and molecular dynamic simulations to identify the key steps regulating lipid movement through the groove. Lipid scrambling is limited by two constrictions defined by evolutionarily conserved charged and polar residues, one extracellular and the other near the membrane mid-point. The region between these constrictions is inaccessible to lipids and water molecules, suggesting that the groove is in a non-conductive conformation. A sequence of lipid-triggered reorganizations of interactions between these residues and the permeating lipids propagates from the extracellular entryway to the central constriction, allowing the groove to open and coordinate the headgroups of transiting lipids.

[1] Department of Anesthesiology, Weill Cornell Medical College, 1300 York Avenue, New York, NY 10065, USA. [2] Department of Physiology and Biophysics, Weill Cornell Medical College, 1300 York Avenue, New York, NY 10065, USA. [3] Institute for Computational Biomedicine, Weill Cornell Medical College, 1300 York Avenue, New York, NY 10065, USA. [4] Department of Biochemistry, Weill Cornell Medical College, 1300 York Avenue, New York, NY 10065, USA. [5] Present address: Korea Brain Research Institute (KBRI), Daegu, Republic of Korea 41068. Correspondence and requests for materials should be addressed to H.W. (email: haw2002@physbio-tech.net) or to A.A. (email: ala2022@med.cornell.edu)

Biological membranes, composed of a double layer of lipids, define the boundaries of living cells and their sub-compartments. In eukaryotes, the outer and inner leaflets of the plasma membrane differ in their lipid compositions, with the outer leaflet containing mostly phosphatidylcholine (PC) and sphingolipids, while the inner leaflet is composed of phosphatidylethanolamine (PE), the negatively charged phosphatidylserine (PS), and phosphoinositides[1]. Two classes of ATP-driven pumps, flippases and floppases[2–4], create and maintain the membrane asymmetry, whose rapid collapse occurs during a process called phospholipid scrambling that externalizes PS, an effector in numerous cell activities such as blood coagulation and the phagocytic clearance of apoptotic cells[5–10]. The proteins mediating this process are called phospholipid scramblases and their identity had remained controversial until recently when members of two unrelated families of integral membrane proteins, the TMEM16[11–17] and Xk-related (Xkr)[18–20] proteins, were identified as regulated phospholipid scramblases that respond to different physiological stimuli. Increases in cytosolic $Ca^{2+}$ activate the TMEM16s to trigger blood coagulation and membrane repair[11], while caspase cleavage is required for Xkr activation during apoptosis[18]. The TMEM16 scramblases are better understood thanks to extensive structural, functional, and computational investigations[14–17,21–23].

The TMEM16 family comprises dual function $Ca^{2+}$-dependent channel/scramblases and $Ca^{2+}$-activated $Cl^-$ channels (CaCC)[24]. While the two family-founding members, TMEM16A and B, are CaCCs[25–27], it was later realized that the majority of the family members function as $Ca^{2+}$-dependent phospholipid scramblases, which also mediate non-selective ion transport[11,16,17,20,28]. The structures of two TMEM16 homologues, the dual function channel/scramblase nhTMEM16[22] and the CaCC mTMEM16A[23,29], revealed key features of the dimeric TMEM16 architecture (Fig. 1a) and of the $Ca^{2+}$-dependent activation mechanism. Each TMEM16 monomer harbors two $Ca^{2+}$ binding sites and presents a hydrophilic groove-like cavity exposed to the membrane (Fig. 1b). This cavity is lined by five helices, TM3-7, with TM6 and TM7 participating also in the formation of the $Ca^{2+}$ binding site (Fig. 1b). In nhTMEM16 the cavity is wide at its intracellular end, ~35 Å near the inner leaflet (Fig. 1b), and tapers towards the extracellular end to ~5 Å at its narrowest point (Fig. 1b). Recent publications support the idea that this groove serves as the lipid pathway and that scrambling occurs via the credit-card mechanism[2,15,16,21,22,30],

where the lipid headgroups penetrate into the cavity while their hydrophobic tails remain embedded in the hydrocarbon core of the membrane (Fig. 1c). Mutations affecting ion transport localize to this region, suggesting that the cavity might serve as a shared conduction pathway for ions and lipids[15,31–33]. In the mTMEM16A channel the narrowing of the cavity at the extracellular end is more pronounced than in the nhTMEM16 scramblase[23], offering a rationale for their functional divergence.

To investigate the mechanisms regulating access and translocation of lipids through the nhTMEM16 groove we combine experimental and computational approaches as described herein. We find that tryptophan mutations in the cavity affect scrambling and channel activity to a similar extent, consistent with the hypothesis that the ion and lipid pathways coincide. High-impact mutations localize to three hot spots: a constriction of the groove near the mid-point of the membrane, the extracellular entrance of the cavity, and residues located between the two constrictions. Our analysis of results from molecular dynamics (MD) simulation of corresponding nhTMEM16 constructs identifies a mechanism by which lipid access to the pathway from the extracellular leaflet is controlled by dynamic rearrangements of a network of interacting polar residues whose dynamical rearrangements entail the elimination of steric constrictions along the groove, thereby enabling lipid scrambling. Extensive mutagenesis experiments corroborate this mechanistic hypothesis. Based on our results we propose a molecular mechanism for lipid permeation through the nhTMEM16 groove. In this model, the extracellular region of the groove interacts with multiple lipid molecules at the extracellular region of the groove to trigger dynamic rearrangements that open the nhTMEM16 groove and allow lipid permeation.

## Results

**Quantification of the scrambling rate constants.** To monitor scrambling in the wild-type (WT) and mutant constructs we used fluorescently-labeled lipid reporters and a dithionite-reduction assay[14,34] (Fig. 2a). Briefly, vesicles are reconstituted in the presence of trace amounts of fluorescently-labeled lipids and the membrane-impermeant reducing agent dithionite is added to the external solution to bleach outer-leaflet fluorophores. Only the outer leaflet lipids are reduced in protein-free liposomes (Fig. 2a), resulting in ~50% loss of fluorescence (Fig. 2c, green

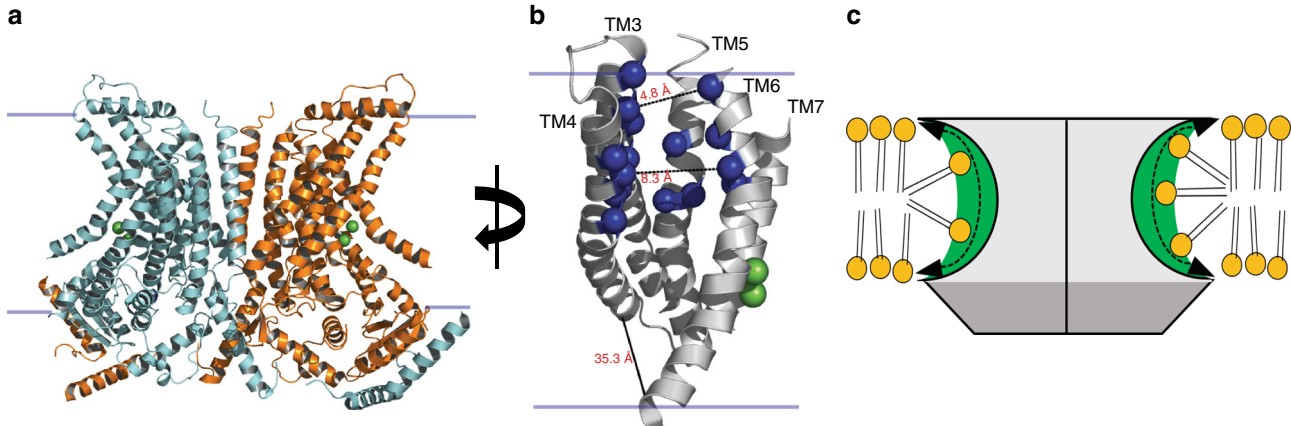

**Fig. 1** Hydrophilic cavity and lipid scrambling of nhTMEM16. **a** The structure of nhTMEM16 viewed from the plane of the membrane. The two monomers are respectively colored in cyan and gold. Green spheres indicated the bound $Ca^{2+}$ ions. Solid lines denote the planes of the membrane. **b** Close up view of the hydrophilic cavity of nhTMEM16. The Cα atoms of the residues mutated to tryptophan are shown as blue spheres and the bound $Ca^{2+}$ ions are indicated as green spheres. Dashed gray lines indicate the distance between the side chains at the extracellular entry (E313/R432), the mid-point and (T333/Y439) at the intracellular vestibule (A356/F463) of the groove. **c** Schematic representation of the credit-card model of lipid scrambling

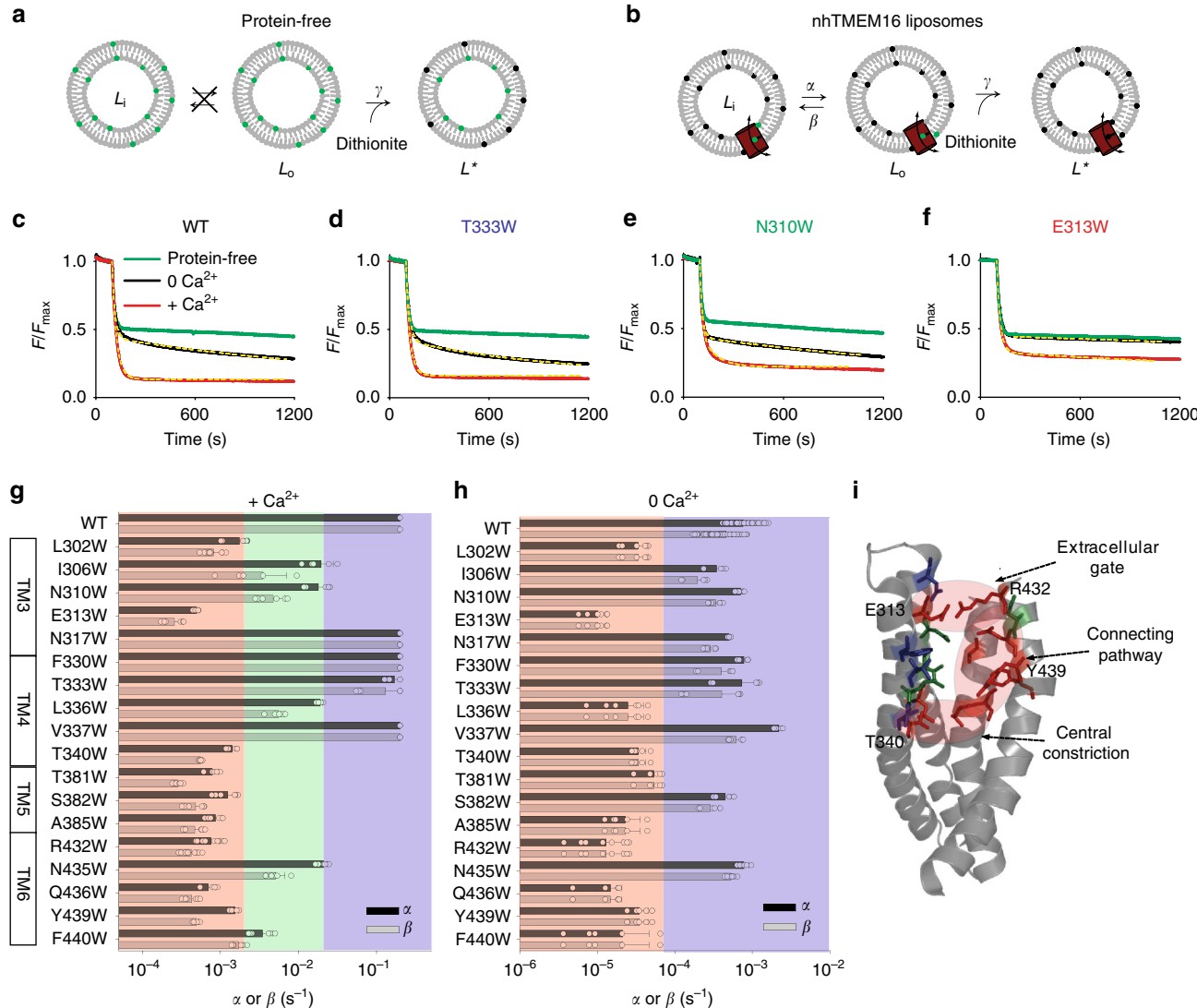

**Fig. 2** Effects of W-scanning on scrambling activity of nhTMEM16. **a**, **b** Schematic representation of the scrambling assay for protein-free vesicles (**a**) and proteoliposomes (**b**). **c–f** Representative traces of the dithionite induced fluorescence decay for WT (**c**) and low- (**d**), medium- (**e**) and high-impact (**f**) tryptophan mutants of nhTMEM16 in the presence (red) and absence (black) of $Ca^{2+}$. Dashed yellow lines indicate fits to Eq. (6). Protein-free traces are shown in green. **g**, **h** Quantification of scrambling rate constants $\alpha$ and $\beta$ for WT and mutant nhTMEM16 in the presence (**g**) and absence of $Ca^{2+}$ (**h**). The effects were categorized into three classes as low- (<10-fold reduction, blue shading), medium- (10 to 100-fold reduction, green shading) and high-impact residues (>100-fold reduction, red shading). Individual data points are shown as empty circles. All data is reported as the mean ± S.D. The number of replicates are indicated in Supplementary Table 1. **i** Mapping of the tested residues on the nhTMEM16 structure according to their impact on scrambling, colors as in **g**, **h**

line). In vesicles containing at least one active scramblase all fluorescently-labeled lipids are eventually bleached (Fig. 2b). Under our reconstitution conditions not all vesicles contain scramblases[14,28], therefore the time course of decay in total fluorescence $F_{tot}$ reflects the behavior of a mixed population of protein-free liposomes and proteoliposomes and is described by Eq. 6 ('Methods')[35] where the only free parameters are the forward and reverse scrambling rate constants, $\alpha$ and $\beta$ (Fig. 2b), and the fraction of protein-free liposomes, $f_0$.

In the presence of $Ca^{2+}$, the kinetics of fluorescence decay to final plateau (Fig. 2c, red line) are similar to those seen in protein-free vesicles, indicating that detection of scrambling is rate-limited by the chemical reduction of the NBD fluorophores. In this case, only a lower limit of $\alpha(+Ca^{2+}) = \beta(+Ca^{2+}) > 0.2\,s^{-1}$ can be determined[35], which corresponds to a lipid transport rate for nhTMEM16 $> 2 \times 10^4$ lipid $s^{-1}$. In the absence of $Ca^{2+}$, scrambling can be directly resolved (Fig. 2c, black line), and

occurs with rate constants $\alpha(0\ Ca^{2+}) \sim \beta(0\ Ca^{2+}) \sim 0.001\,s^{-1}$, corresponding to a transport rate of ~100 lipid $s^{-1}$.

**Probing the lipid pathway of nhTMEM16.** To investigate the role of the nhTMEM16 hydrophilic cavity in ion and lipid permeation we systematically replaced residues lining the groove with tryptophan, expecting the physicochemical properties of its side chain to enable its accommodation in hydrophobic and hydrophilic environments of the groove, and its bulk to hinder passage of lipid headgroups through the groove. The investigations focused on the extracellular side of the groove, as the role of the intracellular vestibule of TMEM16 proteins in lipid scrambling was recently described[15,16,30].

We quantified the impact of 18 substitutions of cavity-lining residues (Fig. 1b) to tryptophan on the scrambling rate constants in the presence and absence of $Ca^{2+}$ using Eq. (6) (Fig. 2d–f,

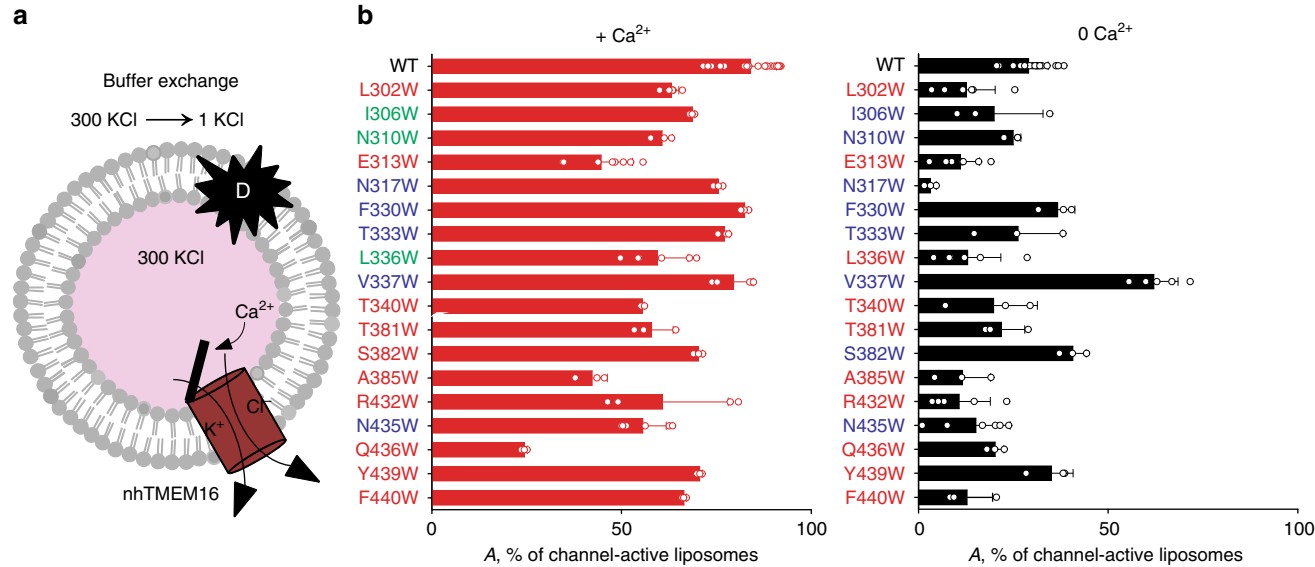

**Fig. 3** Effects of W-scanning on the ion conduction properties of nhTMEM16. **a** Schematic representation of the flux assay. The star with D denotes addition of detergent to dissolve all liposomes. **b** Fraction of the liposomes containing at least one active nhTMEM16 channel in the presence (red) and absence of $Ca^{2+}$ (black). Individual data points are shown as empty circles. All data is reported as the mean ± S.D. The number of replicates are indicated in Supplementary Table 1

Supplementary Figure 1). Because rate constants of the mutants span ~3 orders of magnitude, from ~$10^{-4}\,s^{-1}$ to >$0.2\,s^{-1}$, clustering into classes of 10-fold change produced three groups in the presence of $Ca^{2+}$, and two in its absence (Fig. 2g, h). Changes in the scrambling rate constants reflect specific effects of the mutations on the activity of the protein; all constructs yield well-folded proteins that elute as single mono-disperse peaks on gel filtration and incorporate into liposomes with WT-like efficiency (Supplementary Figure 2). Calculated rate constants are macroscopic parameters that reflect the number of active molecules per vesicle, their open probability, and unitary conductance. Our analysis cannot reveal whether a given mutant affects one, two or all factors. However, a marked reduction in the plateau fluorescence level indicates that a higher fraction of liposomes contains no active proteins[14,36], suggesting that some mutations might favor a long-lived inactivated state of the scramblase. Finally, the decreased scrambling rate did not reflect a reduced affinity for $Ca^{2+}$, as addition of 3 mM $Ca^{2+}$ did not accelerate scrambling (Supplementary Figure 3).

Most mutations in TM3 and TM4 (N317, F330, T333, and V337) have low to no impact on scrambling (<10-fold change in the rate constants) in the presence and absence of $Ca^{2+}$ (Fig. 2g, h). An exception is V337W, with ~2-fold increased scrambling rate constant in the absence of $Ca^{2+}$ (Fig. 2h). Detection in the presence of $Ca^{2+}$ is rate-limited by the chemical step, so we cannot resolve a possible increase in scrambling. Three residues located in the inner part of the groove, two on TM3 (I306, N310), and one on TM4 (L336), significantly impair scrambling, with a 10- to 100-fold reduction in the rate of lipid transport in the presence of $Ca^{2+}$ (Fig. 2g). The largest reduction in lipid transport, >100-fold (Fig. 2g), was found for residues that localize to three structurally distinct areas: a constriction of the groove near the mid-point of the membrane formed by L302, T340, T381, and S382, the extracellular entrance of the cavity formed by E313 and R432, and four residues (A385, Q436, Y439, and F440) on TM5 and 6 between the two constrictions (Fig. 2i). Overall, the distribution of the high-impact residues within the lipid cavity is distinctly asymmetric: in TM5 and TM6 each residue mutation tested severely affects scrambling, whereas in TM3 and TM4 the only two high-impact positions are at contact points with TM5

and TM6 (Fig. 2i). These results are in general agreement with recently published work[15], except for the T333W mutation that has minimal impact in reconstituted vesicles while the T333V substitution impairs scrambling in cells. Further work will clarify whether this one discrepancy reflects differences in the assays or in the introduced side chains, but our results support the hypothesis that the membrane-exposed hydrophilic cavity of nhTMEM16 serves as the lipid pathway for scrambling, and identify two constriction points that are likely rate-limiting for lipid transport.

**Identification of residues essential for ion permeation**. We showed previously that nhTMEM16, like afTMEM16 and TMEM16F, also mediates $Ca^{2+}$-dependent non-selective ion transport[14–17,28]. To quantify the fraction of liposomes containing at least one active channel, $A(\pm Ca^{2+})$, we used an ion flux end-point assay (Fig. 3a)[14,28]. In the presence of $Ca^{2+}$, $A_{WT}(+Ca^{2+})$ is ~85% as most vesicles contain at least one active copy of WT nhTMEM16 (Fig. 3b), while in the absence of $Ca^{2+}$ $A_{WT}(0\ Ca^{2+})$ drops to ~30% (Fig. 3b). The non-null value of $A_{WT}(0\ Ca^{2+})$ is consistent with the nhTMEM16 channel retaining a low baseline activity even in the absence of $Ca^{2+}$[14,28]. These flux experiments while valuable, provide only limited insights, as they are end-point measurements that lack kinetic resolution and overestimate the ion throughput of the reconstituted channels for two reasons. First, each liposome contains on average 5–6 copies of the nhTMEM16 channel. Second, exchange of the extracellular solution occurs over ~90–120 s, but openings of any of the channels for <1 s are sufficient to dissipate the ~$10^5$–$10^6$ ions reconstituted in each vesicle. Thus, we can estimate that a significant reduction (>2-fold) in the fraction of liposomes containing at least one active channel corresponds to a >100-fold reduction in the ion throughput of the channel. This implies that the assay underestimates the effects of the mutations. It is thus remarkable that three mutations (E313W, A385W, and Q436W) reduce $A(+Ca^{2+})$ by >50% compared to the WT level (Fig. 3b). In the absence of $Ca^{2+}$, six mutants (E313W, N317W, L336W, A385W, R432W, F440W) reduce $A(0\ Ca^{2+})$ by >50% (Fig. 3b). Notably, V337W is the only mutation that increases the

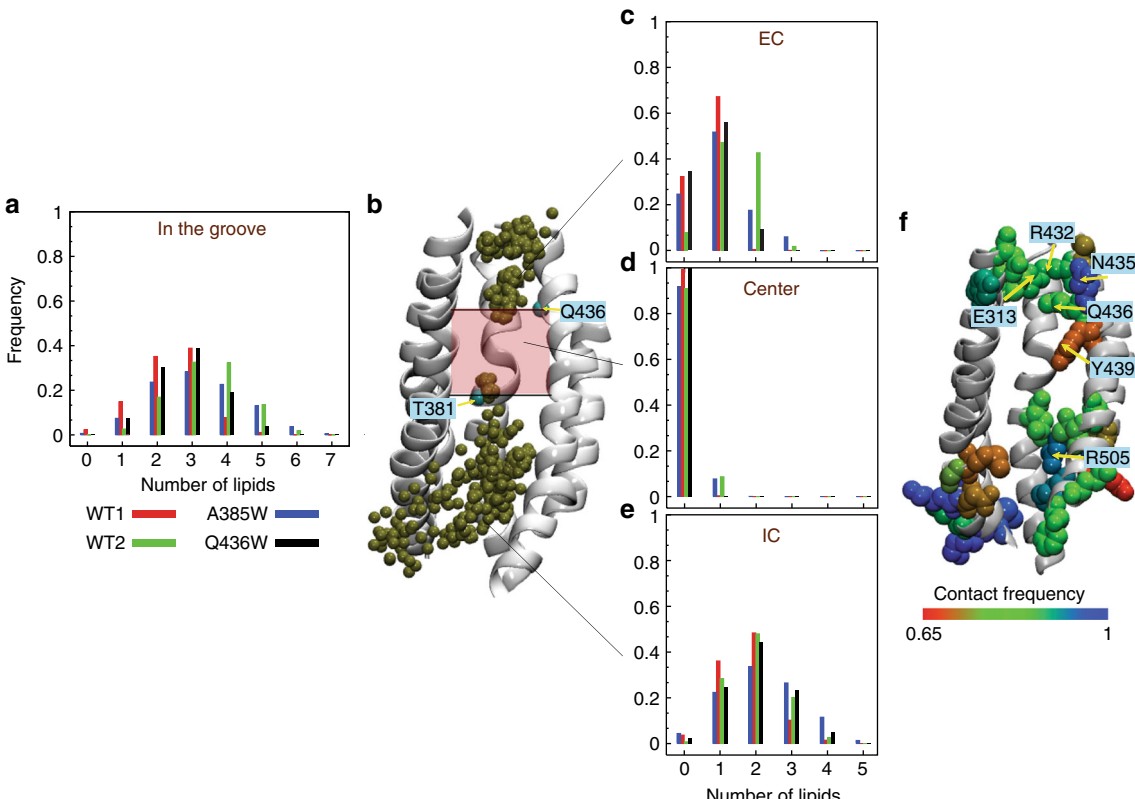

**Fig. 4** Lipid headgroups transiently populate the hydrophilic groove of nhTMEM16. **a** The fraction of time (i.e., fraction of trajectory frames) a specific number of lipid headgroups simultaneously occupy the groove in the WT1 (red), WT2 (green), A385W (blue), and Q436W (black) simulations (see 'Methods' for the definition of the groove and its different regions). For each construct, the data shown is the average over the two protomers of nhTMEM16. **b** Snapshot of the nhTMEM16 (TM helices 3–7) from WT2 simulation illustrating coverage of the groove by lipid phosphorus atoms. Gold-colored spheres represent superposition of the phosphorus atom locations from the entire 4 μs-long MD trajectory using a stride of 30 ns time-interval between the trajectory frames. The location of the central region of the groove devoid of lipids is indicated by the shaded rectangle between the thick black lines that indicate as well the z-axis positions of the Cα atoms of residues T381 and Q436 (cyan spheres). **c–e** Same as in **a** only measured separately for three different compartments of the groove: EC side of the groove (in **c**), center of the groove (**d**), and IC side of the groove (**e**). **f** nhTMEM16 structure (TMs 3–7) showing residues of the groove (in van der Waals and colored) that are in contact with lipid headgroup >65% of the time in the combined WT1 and WT2 trajectories (the data presented is the average over the two protomers of nhTMEM16). A lipid headgroup was considered in contact with a residue if the phosphorus atom of the lipid was within 7 Å of any atom of the residue (center-to-center distance was considered)

channel activity of nhTMEM16 in the absence of $Ca^{2+}$ by >2-fold (Fig. 3b), mirroring its effect on lipid scrambling (Fig. 2g, h). Overall, the mutations that affect ion channel activity mostly overlap with those that influence lipid scrambling: the four residues with 'low' impact on scrambling have WT-like channel activity in $+Ca^{2+}$ and the four with 'medium' scrambling impact have little effect channel activity. Of the ten loci with high impact on scrambling, two (A385 and Q436) also significantly reduce channel activity, while five (T381, S382, R432, Y439, and F440) have little to no effect on ion transport. The low to intermediate effects of the other three residues are likely underestimates due to the limitations of the flux assay. In the absence of $Ca^{2+}$ the pattern is similar, with the exception of N317W which has no effect on scrambling but reduces $A_{N317W}(0\ Ca^{2+})$ ~7-fold. Our findings suggest that the same residues play key roles in both transport processes and are consistent with the idea that ions and lipids share a common permeation pathway formed by the hydrophilic groove[15,23,33].

**A barrier to lipid movement in the mid portion of the groove**. To develop a mechanistic model of lipid scrambling we used atomistic MD simulations of nhTMEM16 embedded in 3:1 POPE:POPG or in POPC lipid membranes (Supplementary

Table 2, 'Methods'). To learn how the pathway for lipid translocation is formed and regulated we quantitatively analyzed multiple replicas of long MD runs and examined the dynamic characteristics of the groove in the WT and two mutant constructs that significantly impair scrambling (A385W and Q436W, Fig. 2i).

Analyzing two independent simulation trajectories of WT nhTMEM16 (WT1 and WT2, of 3 and 4 μs respectively), and one each for the A385W and Q436W constructs (2 μs each) in POPE:POPG membranes, we observed penetration of multiple lipids into the hydrophilic groove of the protein in all, and with comparable average lipid occupancy (Fig. 4a, Supplementary Figures 4a and 4e). Notably, the lipid distribution within the groove is discontinuous (Fig. 4b–e, Supplementary Figures 4b-d and 4e-h): the headgroups mostly localize in the wide intracellular (IC) portion of the groove (Fig. 4e) and in the extracellular (EC) region of the pathway (Fig. 4c). The mid portion of the groove, delimited by T381 and Q436 (Fig. 4d), is nearly devoid of lipids (a low probability of at most single occupancy). This lipid-depleted region is the narrowest in the groove and is also poorly hydrated, as waters cluster to the intra-cellular and extra-cellular vestibules (Supplementary Figure 5).

Over the course of multiple trajectories lipid headgroups were observed to engage with specific sets of residues (Fig. 4f,

Supplementary Figure 6), such as R505 in the intracellular vestibule (Fig. 4f). At the extracellular end, several polar residues form extensive contacts with the penetrating lipids (Fig. 4f, Supplementary Figure 6). Notable among these, are several of the positions we identified as high impact in the Trp scanning mutagenesis experiments, such as E313, R432, Q436, and Y439 (Fig. 2g–i). To understand how these residues interact with lipids and how they participate in the formation of a lipid-permissive translocation pathway we followed in the MD trajectories the dynamics of the headgroups monitoring the evolution of pairwise distances to these polar residues in conjunction with structural rearrangements of the protein.

**Lipid translocation requires opening of the pathway.** Our MD simulations show that lipid headgroups approaching the EC end of the pathway trigger a rearrangement of a triad of charged residues (E313, E318, and R432) which gradually eliminates the steric barriers delimiting the lipid-depleted and water-depleted zone and thus enabling lipid transfer between leaflets. The dynamic nature of the interactions within the E313/E318/R432 triad is revealed by the destabilization and rearrangement of the electrostatic interaction between E313 and R432 by lipids (Supplementary Figure 7, red traces). In all MD trajectories when the R432 sidechain leaves the pair-interaction with E313 and engages with E318, one helical turn above on TM3 (Supplementary Figure 7, black traces), the number of lipid headgroups surrounding E313/R432 increases (Supplementary Figure 8, 9). Both POPE and POPG lipids contact the charged pair, with at most one PG headgroup involved in the interactions at any time (Supplementary Figure 9). The near-linear relationship between $f_{int}$ (defined as the fraction of time when the alternative, E318-R432, bond is formed and the E313-R432 interaction is disrupted), and $f_{lip}$ (defined as the fraction of time in which two or more lipid headgroups are present near E313/R432) (Fig. 5), suggests that the lipid interactions modulate the stability of the interaction network. Exceptions to this pattern are observed in the WT1 trajectory and in one protomer of the A385W trajectory (Fig. 5, red and blue circles denoted by *). In both cases a rare mode of lipid penetration occurs, where the lipid inserts with both the headgroup and tail into the groove (Supplementary Figure 10a, b), plugging the pathway and causing its dehydration. Similar rare events of lipid transfer path blockade by an inserted lipid tail were observed in a recent study of lipid scrambling by the G protein-coupled receptor opsin[37].

The two lipids that promote the switch of R432 interaction from E313 to E318 differ in their engagement with the charged residues (Fig. 6). One interacts with R432 from the side of the groove (Fig. 6a, purple lipid), while the other inserts directly into the groove near E313, occupying the space vacated when the side-chain of R432 interacts with E318 (Fig. 6b, green lipid). The structural rearrangements of the triad, and the ensuing increase of protein–lipid interaction frequency are observed in both monomers of the WT2 trajectory (Supplementary Figure 7b, black and red traces; Supplementary Figure 11), but lipid permeation does not progress beyond these initial stages. Such progress is observed in the non-blocked protomer of the A385W mutant (Fig. 6c, d) where the lipid reaches the intracellular leaflet. The first step is disengagement of R432 from both E313 and E318 (Fig. 6e, time-point denoted by C), while it remains coordinated by lipids (Fig. 6c, green and purple lipids; Fig. 6f). This allows TM3 and TM6 to move apart, as several polar interactions between them are disrupted, and the extracellular vestibule of the groove widens by ~3 Å with Q436 and E313 moving away from each other (Fig. 6e, green trace). The central constriction is eliminated and opens by ~6 Å as the Y439 side

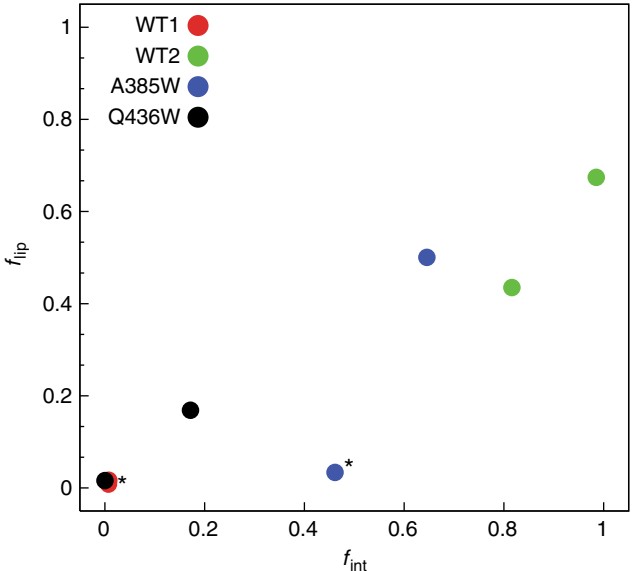

**Fig. 5** Switching of R432 from E313 to E318 promotes lipid accumulation near the extracellular side of the groove. Frequency of finding two or more lipids ($f_{lip}$) vs frequency of the protein configuration in which E313-R432 bond is broken and E318-R432 is formed ($f_{int}$). $f_{int}$ is the calculated fraction of time when the E318-R432 bond was formed (defined by a minimum distance of 3 Å or shorter between these two residues), and the E313-R432 interaction was disrupted (defined by a minimum distance larger than 3 Å between these two residues); $f_{lip}$ is the calculated fraction of time in which two or more lipid headgroups are present near the E313/R432 pair of residues. Each symbol represents a data point for one of the protomers of nhTMEM for a particular construct. Close to linear relationship between $f_{lip}$ and $f_{int}$ is inferred from the linear regression of the data which yielded Pearson correlation coefficient of 0.6. A star denotes the data for one of the protomers in the A385W and WT1 constructs with a rare mode of lipid binding in the groove in which lipid tail penetration to the extracellular side of the groove was observed (Supplementary Figure 10)

chain rotates away from T333 and towards the membrane (Fig. 6e, blue trace; Fig. 6g). These dynamic rearrangements generate an opening into which a third lipid partitions from the external leaflet (Fig. 6c, blue lipid; Supplementary Figure 12) and proceeds to diffuse through the opened groove to the IC leaflet. The flip is completed as the permeating headgroup reaches ~10 Å below the mid-point of the membrane and its tails are directed towards the EC leaflet (Fig. 6i, time-point denoted D; Supplementary Figure 13). This sequence of rearrangements brings to light the essential role of the Y439 side chain that coordinates the headgroup of the flipped lipid throughout the permeation event (Fig. 6h). Thus, lipid translocation through the groove requires a time-ordered sequence of lipid-dependent local conformational changes in the network of interactions among the polar residues at the extracellular side of the groove, E313, E318, R432, Q436, and Y439, which (i)-is triggered by interactions with individual lipids, and (ii)-gates the lipid translocation pathway.

Because the switch of molecular interactions and conformational rearrangements of the EC gate are rare events, even in the extended time scales of the MD simulations described above, we increased sampling of such rare events by implementing an iterative protocol we described recently for analyzing scrambling in a GPCR[37]. Multiple replicas of unbiased MD simulations are run in sequential stages, so that initiation of one stage is informed by output from the preceding one ('Methods', Supplementary Figure 13a). Using this approach, we generated 24 statistically

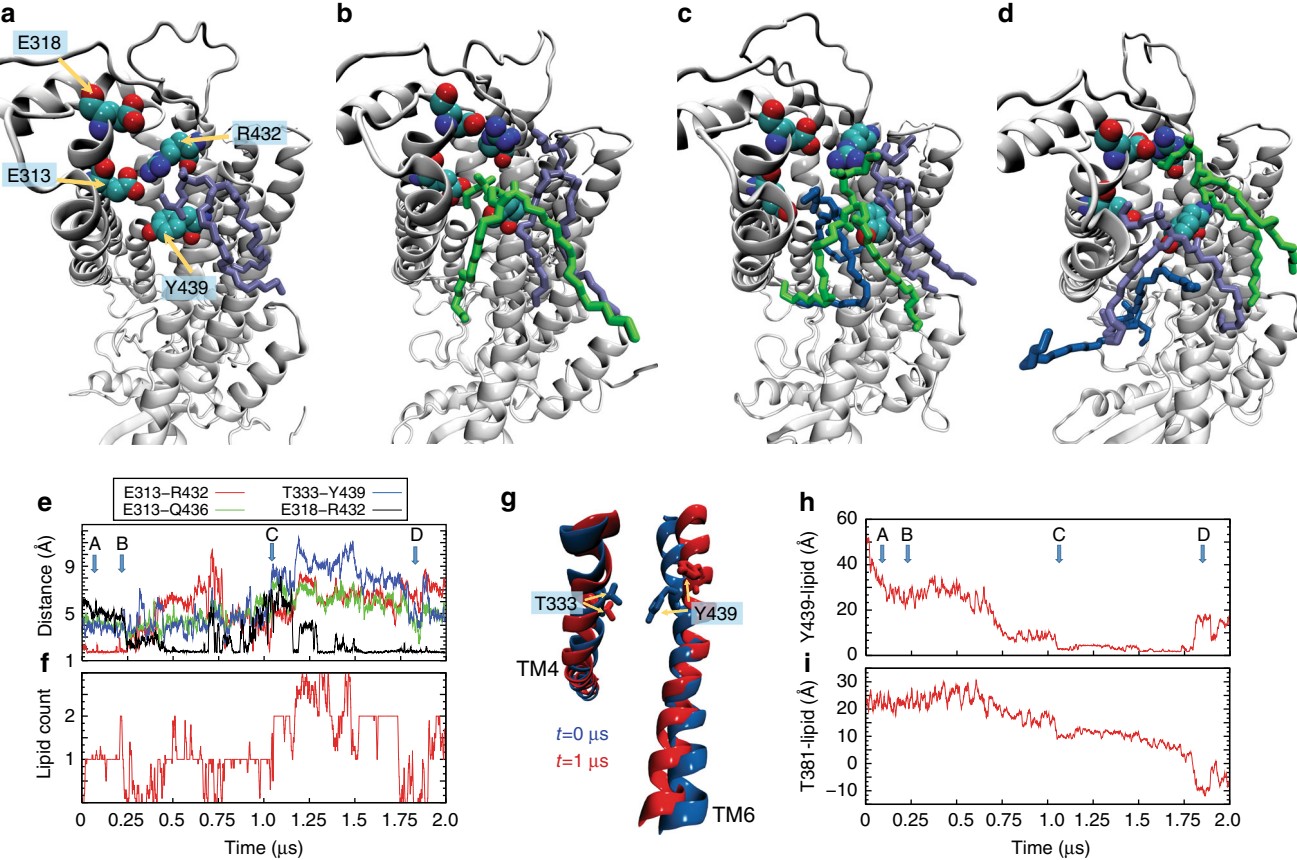

**Fig. 6** Rearrangements of the groove underlying phospholipid flipping. **a–d** Positioning of residues E313, E318, R432, and Y439 and of lipids neighboring the E313/R432 pair at various stages of the simulations. **a** The E313-R432 gate is closed; **b** R432 switches interaction partner from E313 to E318; **c** R432 breaks away from both E313 and E318 allowing lipid penetration deep into the groove (blue lipid); **d** the translocated lipid (in blue) completes the flip as its headgroup is on the level of the lipid headgroups in the intracellular leaflet. The time points at which these snapshots were taken from the A385W trajectory are indicated by arrows in **e**, **h**. **e** Time evolution of the distances between E313-R432 (red), E313-Q436 (green), T333-Y439 (blue), and E318-R432 (black). **f** Time evolution of the number of lipid headgroups within 5 Å of residues E313 and R432. **g** Snapshots at $t = 0\,\mu s$ (blue) and $t = 1\,\mu s$ (red) time-points along the A385W simulation showing conformational switch in the T333 and Y439 residues (in licorice and labeled) and corresponding rearrangement of the TM4 and TM6 helices (in cartoon). Note the outward movement of the Y439 side-chain and concomitant repositioning of the extracellular end of the TM6 that opens a pathway for lipid translocation. **h** Time evolution of the distance between the headgroup of the flipped lipid and the Y439 side chain (**h**) and of the z-axis position distance between the penetrating lipids and the Cα atom of the T381 residue (**i**)

independent trajectories of WT nhTMEM16 in POPC membranes (WT^ensemble in Supplementary Table 2) and observed complete lipid transfer events between the inner and outer leaflets in 7/24 simulations (Supplementary Figure 13b, g). We used the tICA dimensionality reduction approach[37–42] to analyze the involvement of the gate residues in lipid translocation. The reduced tICA space onto which the full complement of Stage 3 trajectories was projected, was constructed from eight dynamic variables (Supplementary Table 3) that describe: (i) the position of the scrambled lipid along the groove; and (ii) interactions between the gate residues during the translocation process ('Methods'). The space is described by the first two tICA vectors (Fig. 7a), which represent ~90% of the total dynamics of the system (Supplementary Figure 14a). Following established procedures[42], the ensemble of conformations represented in this 2D tICA space was discretized into 50 microstates using automated clustering k-means algorithm for structural analyses (Supplementary Figure 14b). These microstates cover the configurational space of the entire system as lipid translocation occurs from the IC to EC leaflets. A systematic comparison of the structural characteristics of these states to those observed in our long MD simulations is detailed in the Supplementary Notes. Remarkably, the tICA analysis identifies all key mechanistic states

outlined above for the lipid translocation from the EC-leaflet to the IC-leaflet, but in reverse order. Specifically, we find that the EC gate undergoes the same conformational steps during lipid translocation (Fig. 7, Supplementary Figure 15) and that simultaneous destabilization of the interactions tethering TM3 to TM6 (between E313-R432, E318-R432 and T333-T439) is required to release the flipped lipid into the external leaflet (Supplementary Figure 15c, Microstates 4–6). Additionally, opening of the E313-R432 and E318-R432 gates involves an increase in numbers of lipid headgroups and water molecules in the EC vestibule (Supplementary Figure 15b, Microstates 4–6). Finally, the tICA analysis also identifies rare events, such as those where the groove is blocked by the lipid tail (Fig. 7a, denoted by *; Supplementary Figure 10).

**Analysis of Trp mutations stabilizing a closed EC gate.** The sequence of steps describing the involvement the EC portions of TM3 and TM6 in the scrambling mechanism is supported by the results of the mutagenesis experiments in which these regions emerged as critical for efficient lipid scrambling (Fig. 2g, h). Analysis of simulations of the Q436W and A385W mutants further substantiates our proposed gating mechanism. We find

that the W436 side chain forms a long-lived hydrogen bond with E313 (Supplementary Figure 7d, panel Q436W) that locks TM3 and TM6 into interactions favoring the closed conformation of the EC of the groove. Similarly, H-bonding between W385 and E313 is evident in simulations of the A385W protomer in which the pathway is closed by the rare mode of lipid binding (Supplementary Figure 16a, c). In contrast, this H-bond is not stable in the A385W protomer in which lipid transfer occurs (Fig. 6; Supplementary Figure 16b, d).

**Probing the mechanistic steps in lipid pathway opening.** The combined experimental and computational analysis described thus far suggested a molecular mechanism for lipid scrambling. To test our proposal we designed and functionally tested several additional mutants of nhTMEM16. First, we varied the physico-chemical characteristics of the 313 and 432 side chains to probe the role of charged side chains at these locations. Charge-neutralizing mutations (E313Q, R432L, and R432Q), or charge reversal (E313R and R432E) (Fig. 8, Supplementary Figure 17),

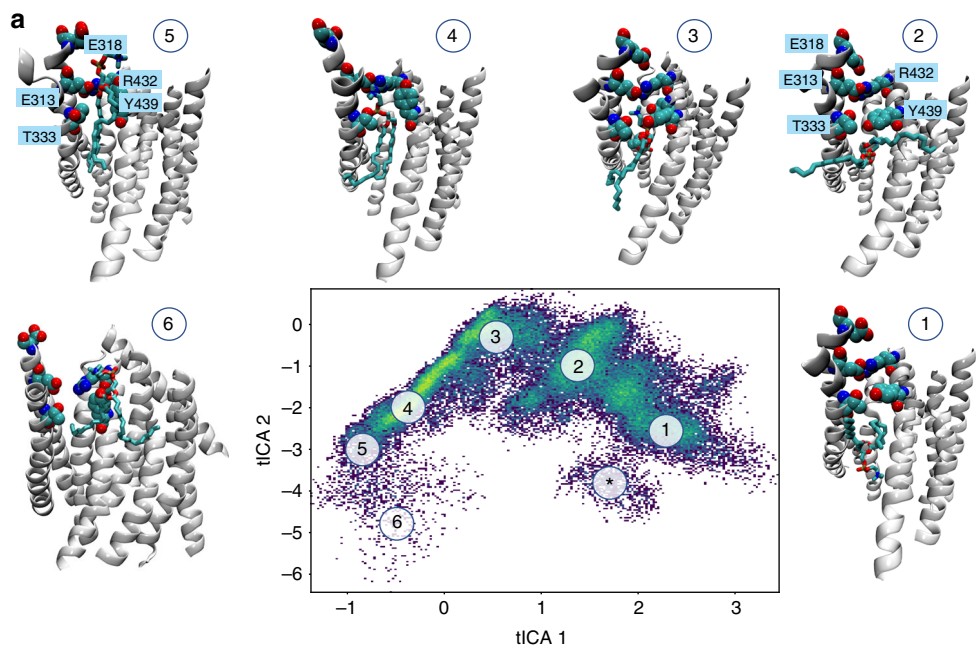

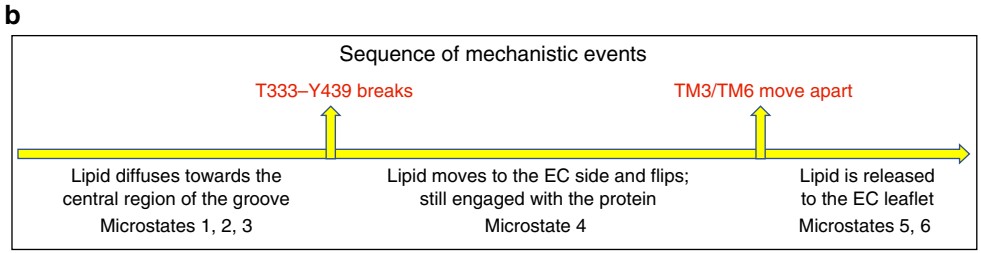

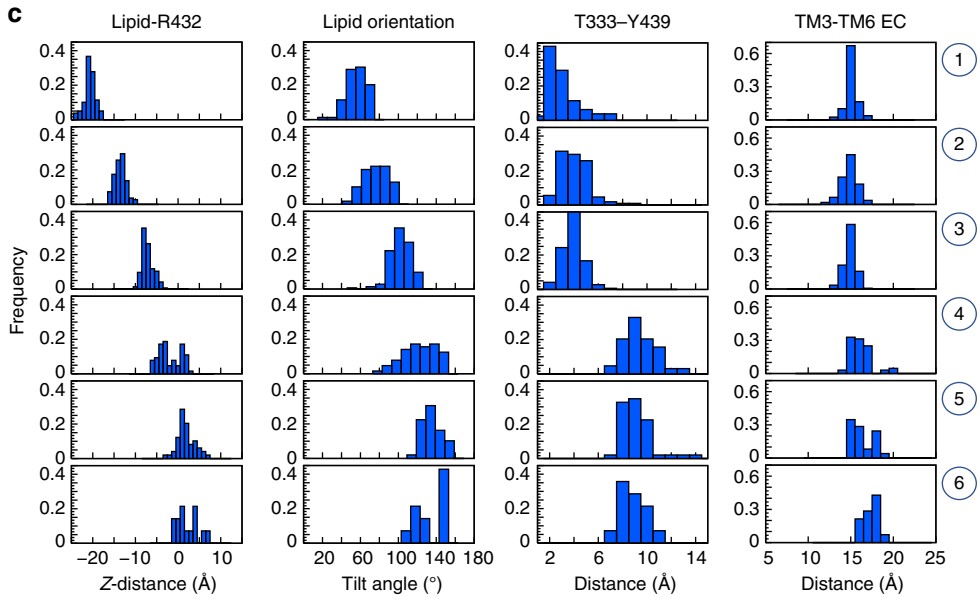

reduce lipid transport by >100-fold (Fig. 8g). Similarly, alanine substitutions at either position reduce scrambling by >100-fold compared to the WT (Fig. 8g). The only functionally tolerated substitutions are charge-conserving mutations, E313D and R432K, which exhibit WT-like characteristics (Fig. 8c, f, g). These results support our proposal that the charges at E313 and R432 are essential components of the dynamic interacting network that regulates the opening of the extracellular side of the groove. Intriguingly, channel activity responded differently to these mutations, as most constructs maintained significant $Ca^{2+}$-dependent ion channel activity in the fluxes, with $A(+Ca^{2+}) >$ 50% (Fig. 8i). The only exception was the charge reversal mutant E313R whose activity was impaired and $Ca^{2+}$-independent (Fig. 8i).

Next, we probed the roles of the other residues comprising the polar network described above, E318, Q436, and Y439. Replacing E318 with Alanine or Arginine (E318A, E318R) reduced the scrambling rate constants respectively by 40-fold and 100-fold (Fig. 9a, g, h). In contrast, the polarity-preserving E318Q mutation reduced scrambling only ~2–3 fold (Fig. 9b, g, h), likely reflecting a more limited ability to engage with R432 compared to the native E318. The effects of these substitutions are consistent with a decreased ability of mutants to disrupt the R432-E313 interaction, thus favoring the closed conformation of the pathway. We probed the R432/E313/E318 interaction network by combining mutations at these positions. The charge swap mutant, E313R/R432E, which preserves the electrostatic interaction but not the possibility of the R432 switch to E318, reduces scrambling by >100-fold (Fig. 9g, h). Similarly, simultaneous mutation of both or all three residues to Ala, E313A/R432A and E313A/E318R/R432A, severely impairs activity (Fig. 9c, g, h). Finally, reversal of the charges in this network, E313R/E318R/R432E, results in a completely inactive protein (Fig. 9c, g, h). These results suggest that a static interaction network is not sufficient for function. Rather, our data support the hypothesis that the integrity and remodeling of the interaction network during gating is essential for scrambling.

We mutated Q346 and Y439 to Alanine to probe their role in scrambling. If the groove is a passive conduit for lipid permeation, these mutations should have a lower impact than the previously analyzed Trp substitutions (Fig. 2). In contrast, the Q436A and Y439A mutations slow down scrambling ~100-fold (Fig. 9g, h), consistent with our proposal that the Y439 side chain is critical for the coordination of the lipid headgroup during translocation (Fig. 6g, i, Fig. 7) and that Q436 is important for the stabilization of the R432-E313 interaction.

## Discussion

The TMEM16 $Ca^{2+}$-dependent scramblases and channels form a membrane-exposed hydrophilic groove[23,43] that serves as the lipid permeation track in a credit card mechanism[2,43]. Several studies support this hypothesis and identified residues important for this process[15,16,21,30]. However, these studies viewed the groove as a passive conduit for lipid transfer and the role of its dynamic rearrangements during scrambling was not investigated. Here, our results from a combination of experimental and computational approaches identify two constrictions along the groove that need to be released to allow lipid transfer between the membrane leaflets. We propose, and experimentally test, a mechanistic model for the opening of the groove in which coordinated conformational rearrangements of the groove are triggered by lipid interactions at its extracellular vestibule. This leads to a remodeling of the groove and opens a continuous transmembrane pathway for lipid permeation (Fig. 10a–d).

The key findings leading to the proposed mechanistic model are (i)-the identification of the structural basis for the two constrictions along the lipid groove, and (ii)-the modes of lipid interactions with residues at the EC end of the groove that trigger its rearrangement. Opening of the translocation pathway involves the remodeling of a network of interactions of several polar residues. Specifically, we show that three charged residues near the extracellular entryway to the groove on TM3 and TM6 (E313, E318, and R432) are essential for lipid scrambling (Fig. 2). Previous work had shown that E313 and R432 are engaged in an electrostatic interaction and participated in lipid recruitment from the extracellular leaflet to the groove[15,21,22,30]. Our present work expands on these findings by showing that this network controls lipid access to the groove and includes a third charged residue, E318 (Fig. 10a). This network undergoes a dynamic rearrangement in which R432 first switches interaction partner from E313 to E318, and then disengages from both. This breaks the interactions holding together the extracellular portions of TM3 and TM6 (Fig. 10b), allowing TM6 to rotate away from TM3 and inducing opening of the mid-point constriction at Y439 and T333 (Fig. 10c, d). The trigger for the gating mechanism involving the rearrangement of the network charged residues is their association with lipids that bind in specific modes, and the formation of the continuous water and lipid pathway connecting the two sides of the membrane that permits lipid scrambling (Fig. 6). A similar gating mechanism was observed in our recent computational studies of lipid scrambling by opsin[37].

Several lines of evidence support our mechanistic proposal that the E313/R432 pair and its partners form a dynamically rearranging interaction network, rather than the static electrostatic interaction seen in the nhTMEM16 structure[22]. The lack of activity of the charge-reversed mutant E313R/R432E (Fig. 9), which preserves the integrity of the interaction, further argues for additional, position-specific, interactions with neighboring residues. Indeed, all alterations to the number and/or position of the charges within the E313/E318/R432 network severely impair scrambling (Figs. 2, 8, 9), consonant with the proposed mechanism in which the positioning of these charges associated

**Fig. 7** Mechanistic events enabling opening of the pathway for lipid flip from the IC to the EC leaflet. **a** Two-dimensional landscape representing mapping of all Stage 3 trajectories of the WT nhTMEM16 in POPC membrane ($WT^{ensemble}$) onto the first two tICA eigenvectors (tICA 1 and tICA 2, see 'Methods' for details of tICA analysis). The location of microstates 1–6 that represent the translocation of the lipid through nhTMEM16 groove is denoted and the structural representation of these microstates are given in the surrounding snapshots. In these structural models, the advancing lipid is rendered in licorice (colored according to atom type), and relevant groove residues are shown in space fill representations and labeled (in microstates 2 and 5). Microstate denoted by * captures rare mode of protein–lipid interaction in which the hydrocarbon tail of a lipid inserts into the EC side of the groove interfering with the translocation of the advancing lipid (see also Supplementary Figure 10c). **b** Sequence of mechanistic events leading to the lipid flip. **c** Structural characteristics of the selected six microstates. The columns from left to right record the probability distributions of Z-directional distance between the phosphorus atom of the translocated lipid and the $C\alpha$ atom of R432 (the 0 on the x-axis represents position of the lipid with its phosphorus atom aligned in the Z direction with the $C\alpha$ atom of R432), of lipid tilt angle, of minimum distance between T333 and Y439, and of opening between the EC ends of TM3 and TM6 helices measured as the distance between the center-of-masses of two groups: $C_\alpha$ atoms of residues 315 to 318 on TM3, and $C_\alpha$ atoms of residues 432 to 435 on TM6

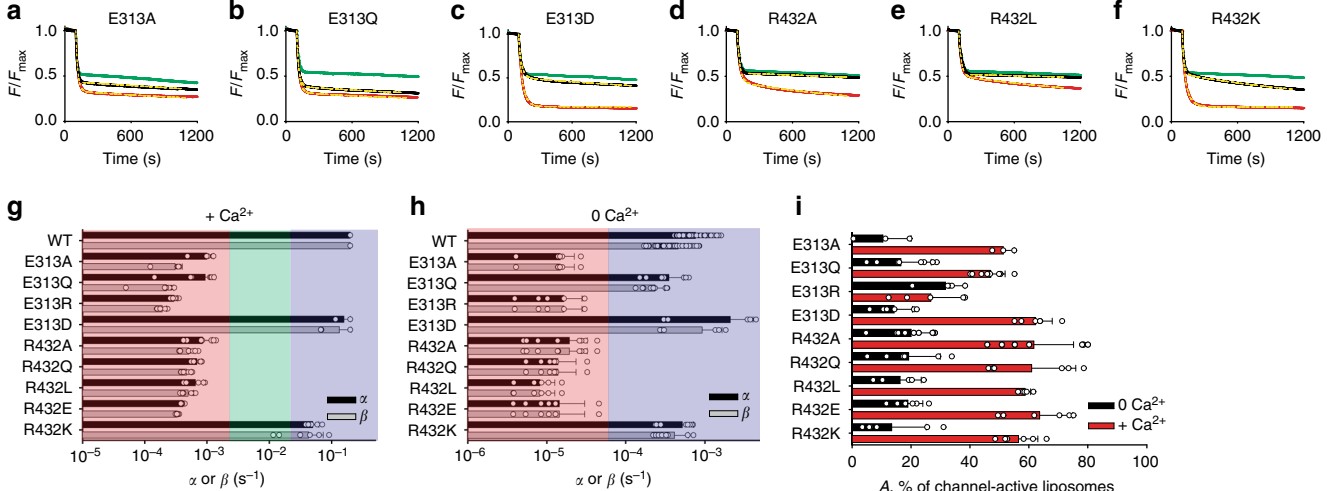

**Fig. 8** The charge of E313 and R432 is essential for lipid scrambling. **a–f** Time course of dithionite-induced fluorescence decay for the indicated nhTMEM16 mutants of E313 and R432 in the presence (red) and absence (black) of $Ca^{2+}$. Dashed yellow lines indicate fits to Eq. (6). Protein-free traces are shown in green. **g, h** Quantification of scrambling rate constants α and β for WT and mutant nhTMEM16 in the presence (**g**) and absence of $Ca^{2+}$ (**h**). The effects were categorized as in Fig. 2g, h. Individual data points are shown as empty circles. **i** Fraction of the liposomes containing at least one active channel in the presence (red) and absence of $Ca^{2+}$ (black). All data is reported as the mean ± S.D. The number of replicates are indicated in Supplementary Table 1

with TMs 3 and TM6 is controlled by more than static pairwise interactions[21]. In addition to forming a lipid binding site, as suggested[21], the modes of interactions of these residues with the headgroups of the permeating lipids, and with Q436, change with the rearrangement of the network (Figs. 5, 6). This rearrangement is essential for the opening and widening of the groove. Interestingly, in the TMEM16A channel-only homologue[23,29], the extracellular vestibule of the groove is narrower than in the nhTMEM16 scramblase, suggesting that rearrangements in this region might be sufficient to allow ion passage but still prevent lipid permeation. In light of this hypothesis, it is interesting to note that the charge reversal mutation R432E retains WT-like channel activity while not conducting lipids (Fig. 8).

Several reports indicate that nhTMEM16 locally deforms the bilayer to allow multiple lipid headgroups to populate its wide intracellular vestibule (Fig. 4)[15,21,44]. Here, we also find that several lipids from the outer leaflet interact with the extracellular entryway of the lipid permeation pathway from positions away from the plane of the outer leaflet (Figs. 4, 6). Such local distortions, together with the widening of the groove that follows the separation of TM3 and TM6, may also allow the TMEM16 scramblases to move lipids with large headgroups between the two leaflets, even without penetration into the groove. This is in good agreement with our recent finding that nhTMEM16 and afTMEM16 transport PE lipids conjugated to polyethylene glycols of molecular weight as high as 5 kDa[35].

In the simulations described here we do not detect ion translocation events, consistent with our finding that nhTMEM16 does not function as an ion channel in a 3:1 POPE:POPG lipid mixture[28]. It is tempting to speculate that the lack of ion translocation, which differs from its observation in otherwise similar simulations[15], may reflect the existence of different modes of opening of the groove that are accessible to lipids only, or lipids and ions, when the membranes surrounding nhTMEM16 differ in lipid composition. Additionally, our simulations were performed without an applied voltage whereas high voltages (reaching 500 mV) were used to promote ion transport in other reports[15]. The voltage drop across the relatively narrow extracellular entryway of the groove is likely to be experienced by the charged E313/E318/R432 triad and provide sufficient energy for the ions, and possibly the lipids, to overcome the hydrophilic lock there. This is

consistent with the finding that scrambling in TMEM16A/16F chimeras entails a slow conformational change of the ion pathway that affects its ion selectivity and pharmacology[15,16].

Our proposed mechanism can naturally account for the otherwise puzzling finding that mutations, far from the extracellular constriction, can convert the channel-only TMEM16A homologue into a scramblase[15,16]. In our view, the groove is no longer a passive conduit for lipid diffusion; rather it is gated at its extracellular entry by a finely tuned network of interactions that regulates the opening of constrictions. Thus, we suggest that in the channel-only TMEM16s[23] the extracellular gate does not open sufficiently to couple to the central constriction, thereby preventing the conformational changes needed to allow lipid scrambling. In contrast, this pathway is functional in the channel/scramblase homologues, so that $Ca^{2+}$-binding can lead to the rearrangements in TM6 required for the opening of the extracellular gate by lipid interactions. We did not explicitly explore the specific elements of $Ca^{2+}$-dependent activation, but we note that TM6 harbors three essential gating elements of the nhTMEM16 scramblase: the $Ca^{2+}$ binding site, the central constriction, and the extracellular polar gate (Fig. 1b). It is therefore tempting to speculate that conformational changes ensuing from $Ca^{2+}$ binding to nhTMEM16 would participate in the opening of the central and extracellular gates via a rearrangement of TM6.

## Methods

**Protein purification.** For the expression of purification of proteins, C-terminal Myc-streptavidin-binding peptide (SBP)-tagged nhTMEM16 construct was used[22]. Expression and purification of WT and mutant proteins were carried out essentially as described[22,28]. *S. cerevisiae* cell (FGY217, from Dr. Drew, University of Stockholm)[45] were transformed with WT and mutant constructs, grown to an O.D. of 0.8 and protein expression was induced by the addition of 2% galactose for 40 h at 25 °C. Cells were resuspended in lysis buffer (150 mM NaCl, 50 mM HEPES, pH 7.6) containing protease inhibitor cocktail and lysed with an EmulsiFlex-C3 homogenizer at above 25,000 psi. Membrane proteins were extracted by supplementing the lysis buffer with 2% n-dodecyl-β-D-maltopyranoside (DDM, Anatrace), and incubated for 1.5 h at 4 °C. Proteins were purified using Streptavidin Plus UltraLink Resin (Pierce) followed by gel filtration chromatography with buffer (150 mM NaCl, 5 mM HEPES pH 7.6, 0.025% DDM) by using a Superdex 200 column (GE Healthcare).

**Reconstitution into liposomes.** For the scrambling assay 0.5 mole % 1-myristoyl-2-{6-[(7-nitro-2-1,3-benzoxadiazol-4-yl)amino]hexanoyl}-*sn*-glycero-3-

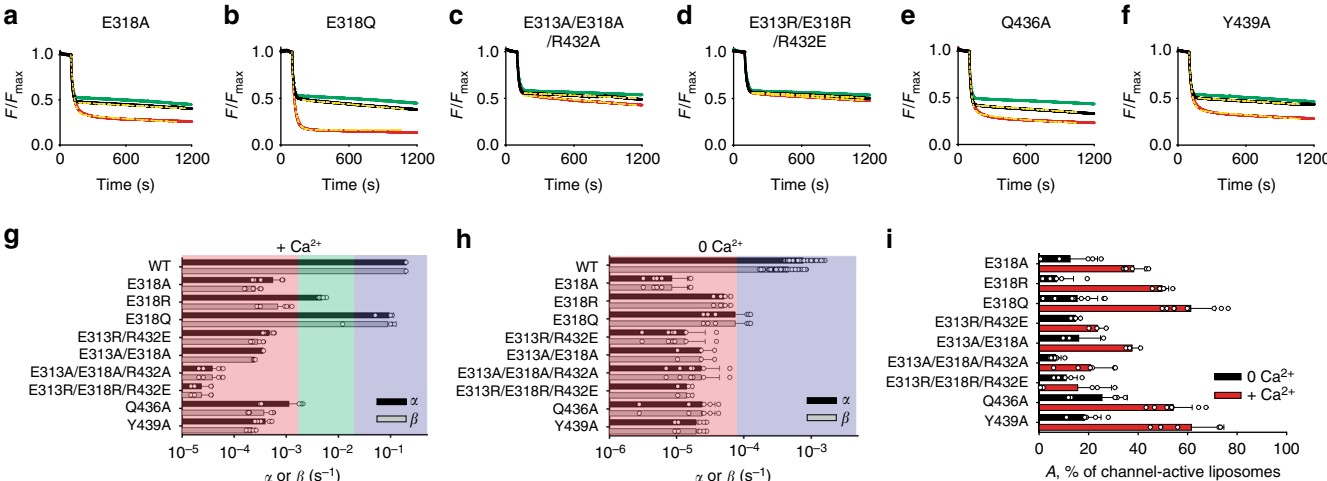

**Fig. 9** Evaluation of the suggested model for lipid scrambling. **a–f** Time course of dithionite-induced fluorescence decay for the indicated nhTMEM16 mutants in the presence (red) and absence (black) of $Ca^{2+}$. Dashed yellow lines indicate fits to Eq. (6). Protein-free traces are shown in green. **g, h** Quantification of scrambling rate constants α and β for WT and mutant nhTMEM16 ($n = 6$) in the presence (**g**) and absence of $Ca^{2+}$ (**h**). The effects were categorized as in Fig. 2g, h. Individual data points are shown as empty circles. Data is mean ± S.D. **i** Fraction of the liposomes containing at least one active channel in the presence (red) and absence of $Ca^{2+}$ (black). All data is reported as the mean ± S.D. The number of replicates are indicated in Supplementary Table 1

phosphoethanolamine (NBD-PE) was additionally added to the lipid mixture. Liposomes were prepared from a 2.25:0.75:1 mixture of 1-palmitoyl-2-oleoyl-*sn*-glycero-3-phosphoethanolamine (POPE): 1-palmitoyl-2-oleoyl-*sn*-glycero-3-phospho-(1′-rac-glycerol) (POPG) and L-α-Phosphatidylcholine (Egg, Chicken-60%). All lipids were purchased from Avanti Polar Lipids. Lipids were dissolved in reconstitution buffer (300 mM KCl, 50 mM HEPES, pH 7.4) in the presence of 35 mM 3-((3-cholamidopropyl) dimethylammonio)-1-propanesulfonate (CHAPS, Pierce). After completely dissolving lipids, the purified protein was added to the desired protein to lipid ratio, typically 5 μg protein/mg of lipid which corresponds to a density of 5–6 proteins per liposome. Proteoliposomes were formed by removing the detergent with four exchange steps of Bio-beads SM-2 adsorbent (160 mg/ml of lipid, Bio-Rad). The $Ca^{2+}$ concentration was adjusted by adding the desired amount of $Ca^{2+}$ or EGTA and equilibrating the liposomes with three freeze/thaw cycles. Liposomes were extruded 21 times through a 400-nm membrane before being used for scrambling and flux assays. To estimate the efficiency of protein incorporation, liposomes were collected after every Bio-beads exchange step and analyzed by SDS-PAGE. Protein amounts were visualized by Coomassie staining and their relative amounts were quantified by densitometry using the Image J software.

**Phospholipid scrambling assay**. Phospholipid scrambling was monitored as described in a previous study[14]. Briefly, 20 μl of extruded liposomes were added into 2 ml of external solution (300 mM KCl, 50 mM HEPES, 0.5 mM $Ca(NO_3)_2$ or 2 mM EGTA, pH 7.4). Fluorescence was acquired in a PTI spectrofluorimeter using excitation and emission wavelengths of 470 and 530 nm, respectively. Forty-microliters of sodium dithionite (40 mM final concentration, added from a stock solution prepared in 1 M Tris) was added to initiate the reaction and bleach the NBD fluorophores.

**Quantification of scrambling activity**. The total fluorescence of a population of nhTMEM16-containing proteoliposomes reconstituted with NBD-labeled lipids is the sum of the signal from vesicles that contain at least one active scramblase, $F_{Scr}(t)$, and of those that are empty, $F_{PF}(t)$, weighted by their relative abundance:

$$F_{tot}(t) = f_0 F_{PF}(t) + (1 - f_0) F_{Scr}(t) \quad (1)$$

where $f_0$ is the fraction of empty vesicles. The fluorescence signal, $F(t)$, is proportional to the sum of the signal from fluorescent lipids in the inner, $L_i(t)$, and outer leaflets, $L_o(t)$, so that

$$F(t) = L_i(t) + L_o(t)$$

In protein-free vesicles only lipids in the outer leaflet are accessible to dithionite, so that the time course of fluorescence decay is described by the following scheme

$$L_i \quad L_o \xrightarrow{\gamma} L^* \quad (2)$$

where $L^*$ is the bleached, non-fluorescent form of the NBD-labeled lipids after dithionite reduction, and the dithionite reduction rate $\gamma = \gamma'[D]$ where $\gamma'$ is the

second order rate constant of dithionite reduction and [D] is the dithionite concentration. Note that the $L_i$ to $L_o$ transition is prohibited as there are no scramblases. Therefore, $F_{PF}(t)$ will be given by

$$F_{PF}(t) = \left( L_i^{PF} + (1 - L_i^{PF}) e^{-\gamma t} \right) \quad (3)$$

with the assumption that $L_i^{PF} + L_o^{PF} = 1$, as the fluorescent lipids are in one of the two leaflets.

The time course of fluorescence decay post-dithionite addition of a liposome containing at least one active scramblase is described by a 3-state Markov model (Fig. 3a)[35,46]

$$L_i \underset{\beta}{\overset{\alpha}{\rightleftharpoons}} L_o \xrightarrow{\gamma} L^* \quad (4)$$

Where α and β are the forward and backwards scrambling rates. The time evolution of this system can be analytically derived[35] and, under the additional assumption that at $t=0$ the system is at the equilibrium state generated by the scramblases, is given by

$$F_{Scr}(t) = \frac{\left\{ \alpha(\lambda_2 + \gamma)(\lambda_1 + \alpha + \beta)e^{\lambda_1 t} + \lambda_1 \beta(\lambda_2 + \alpha + \beta + \gamma)e^{\lambda_2 t} \right\}}{D(\alpha + \beta)} \quad (5)$$

with

$$\lambda_1 = -\frac{(\alpha + \beta + \gamma) - \sqrt{(\alpha + \beta + \gamma)^2 - 4\alpha\gamma}}{2}$$

$$\lambda_2 = -\frac{(\alpha + \beta + \gamma) + \sqrt{(\alpha + \beta + \gamma)^2 - 4\alpha\gamma}}{2}$$

and

$$D = (\lambda_1 + \alpha)(\lambda_2 + \beta + \gamma) - \alpha\beta$$

Substituting Eqs. (3), (5) into Eq. (1) we get the time evolution of the total system is

$$F_{tot}(t) = f_0 \left( L_i^{PF} + (1 - L_i^{PF}) e^{-\gamma t} \right) + \frac{(1 - f_0)}{D(\alpha + \beta)} \left\{ \alpha(\lambda_2 + \gamma)(\lambda_1 + \alpha + \beta)e^{\lambda_1 t} \right.$$
$$\left. + \lambda_1 \beta(\lambda_2 + \alpha + \beta + \gamma)e^{\lambda_2 t} \right\} \quad (6)$$

$L_i^{PF}$ is calculated by fitting of protein-free liposomes and it is fixed during the fitting of WT and mutant traces in the same batch. For all mutant traces we constrained the value of $f_0$ to that determined in WT proteoliposomes in the presence of $Ca^{2+}$ in same reconstitution. This allows proper fitting of the data for low activity mutants, and is supported by the finding that the reconstitution efficiency of all mutants is comparable to that of WT nhTMEM16 (Supplementary Figure 2). It is important to note that in protein-free vesicles we do observe a very slow fluorescence decay (Fig. 2), which likely reflects a slow leakage of dithionite into the vesicles or the spontaneous flipping of the NBD-labeled lipids, whose rate can be estimated with a linear fit to be $L = (5.4 \pm 1.6) \times 10^{-5}$ s$^{-1}$ ($n > 160$). For WT nhTMEM16 and most mutants the leak is >2 orders of magnitude smaller than the rate constant of protein-mediated scrambling and therefore is negligible. However,

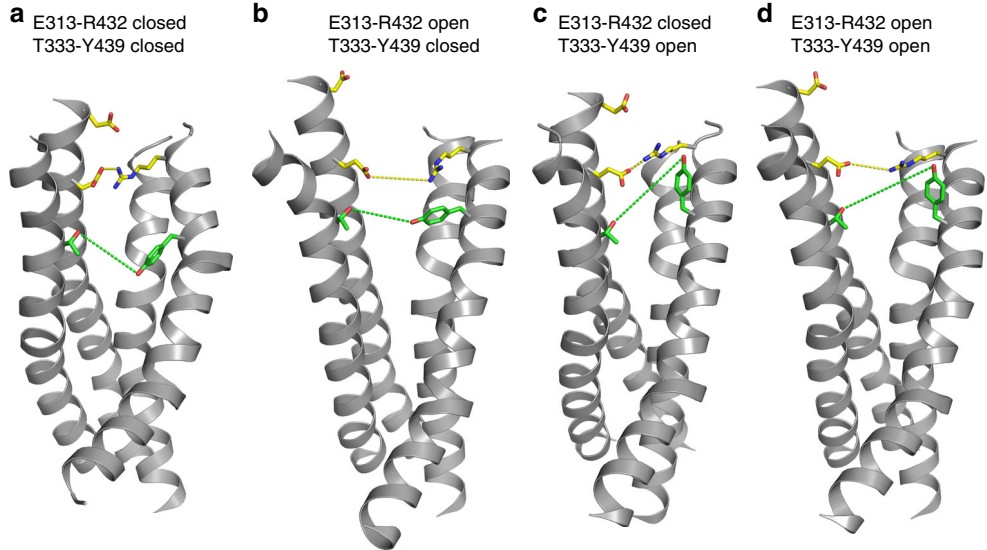

**a** E313-R432 closed
T333-Y439 closed

**b** E313-R432 open
T333-Y439 closed

**c** E313-R432 closed
T333-Y439 open

**d** E313-R432 open
T333-Y439 open

**Fig. 10** Mechanistic steps involved in opening of the extracellular vestibule of the nhTMEM16 groove. **a–d** The nhTMEM16 groove is viewed from the plane of the membrane. For clarity only TM3-6 are shown in ribbon representation, E313, E318, and R432 are shown as sticks in CPK yellow, T333 and Y439 are shown as sticks in CPK green. Dashed lines represent the distance between the oxygen of E313 and the nitrogen of R432 (yellow) or between the oxygen of T333 and that of Y439 (green). **a** The groove is in the conformation from the crystal structure; **b** the extracellular vestibule opens, while the central constriction remains closed; **c** the extracellular vestibule is closed and the central constriction is open; **d** both constrictions are open

some mutants have very low activity, with $\alpha = \beta \ll \gamma$, so that Eq. (6) can be approximated by the sum of a single exponential and a linear function of the form $F_{tot}(t) = A + Be^{-\gamma t} - C(\alpha + L)t$, where $A$, $B$, and $C$ are constants. In these cases, only the linear part of the trace was fit to the linear function

$$F(t) = -C(\alpha + L)t \tag{7}$$

with $L$ fixed to the experimentally determined value of the protein-free liposomes. The values of $\alpha = \beta$ determined in these cases range between $6.5 \times 10^{-6}$ and $7.4 \times 10^{-5}\,\mathrm{s}^{-1}$ validating the use of a linear approximation for the exponential decays. Exact values and number of repeats are reported in Supplementary Table 1.

**Flux assay.** Cl⁻ flux assay was conducted as described previously[14]. In order to exchange the extracellular ionic composition, liposomes were equilibrated in the external buffer (1 mM KCl, 300 mM Na-glutamate, 50 mM HEPES, pH 7.4) by spinning through a Sephadex G50 column (Sigma-Aldrich) pre-equilibrated in external buffer, 0.2 mL of the flow through was added to 1.8 ml of external solution and the total Cl⁻ content of the liposomes was measured using an Ag:AgCl electrode after disruption of the vesicle by addition of 40 μL of 1.5 M n-octyl-β-D-glucopyranoside (Anatrace). The fraction of liposomes containing at least one active nhTMEM16 ion channel, A, was quantified as follows:

$$A = 100\left(1 - \frac{\Delta Cl}{\Delta Cl_{PF}}\right) \tag{8}$$

where $\Delta Cl$ is the change in [Cl⁻] recorded upon detergent addition and $\Delta Cl_{PF}$ is the Cl⁻ content of protein-free liposomes prepared in the presence of 0.5 mM Ca²⁺ and in the same lipid composition.

**Molecular systems.** For all the atomistic MD simulations an X-ray structure of the TMEM16 homologue from *Nectria haematococca* (nhTMEM16, PDBID: 4WIS), was used[22]. Several short fragments missing from the X-ray model (residue segments 1–18, 130–140, 465–482, 586–593, 657–659, 685–691, and 720–735) were reconstructed to complete the protein structure using modeler 9v1[47]. The A385W and Q436W mutations were introduced by BuildModel algorithm available in the FoldX suite[48].

The spatial arrangement of the wild type (WT) and the mutant nhTMEM16 constructs in a lipid bilayer was then optimized using Orientations of Proteins in Membranes (OPM) database[49] and inputted into Membrane Builder module on CHARMM-GUI web-server[50] in order to assemble protein-membrane systems. The protein models were embedded in 1500-lipid size membrane containing 3:1 mixture of POPE and POPG lipids. In additional computational experiments, the WT nhTMEM16 construct was inserted into a POPC bilayer composed of 680 lipids for simulations that more closely match the nhTMEM16-membrane systems investigated in several recently published studies[15,21]. The protein-membrane complexes were then solvated in 0.15 M K + Cl⁻ explicit water solution to achieve electroneutrality.

**Atomistic MD simulation protocols and force-fields.** The assembled molecular systems were first subjected to multi-step equilibration protocol established previously[42] using NAMD software version 2.10[51]. During this stage, the backbone of the protein was first fixed and then harmonically restrained with the constraints on the protein backbone gradually released in three steps of 1 ns each by changing the force constants from 1, to 0.5, and 0.1 kcal/(mol Å2), respectively. This phase was followed by unbiased MD simulations performed with a 2 fs integration time-step, under the NPT ensemble (at $T = 310$ K) and with semi-isotropic pressure coupling, using the Particle-Mesh-Ewald (PME) method for electrostatics[52] and the Nose-Hoover Langevin piston[53] to control the target 1 atm pressure, with Langevin piston period and decay parameters set to 100 and 50 fs, respectively. For the WT system in POPE:POPG membrane, the unbiased MD phase was run in two statistically independent replicates (with new set of velocities generated by random seed) for 370 and 390 ns, respectively. For the mutant constructs in POPE:POPG bilayer, the unbiased MD step was carried out for 23 ns.

After this initial phase, all the molecular systems in POPE:POPG membranes were subjected to microsecond-scale MD simulations (see Supplementary Table 2) on Anton2, a special-purpose supercomputer machine[54]. These production runs were carried out in the NPT ensemble under semi-isotropic pressure coupling conditions (using the Multigrator scheme that employs the Martyna-Tuckerman-Klein (MTK) barostat[55] and the Nosé-Hoover thermostat[56]), at 310 K temperature, with 2.5 fs time-step, and using PME for electrostatic interactions.

The simulations of the WT system in POPC membrane were carried out using a multi-stage iterative protocol (see Supplementary Figure 13a) that was specifically designed to increase sampling of lipid translocation events[37]. Thus, the system equilibration phase described above was followed by 380 ns unbiased MD run (termed Stage 1). The simulations in the next phase (Stage 2) were initiated from the last frame of the Stage 1 trajectory by resetting velocities. The Stage 2 simulations were run in eight statistically independent replicates for 15 ns/each. Stage 3 was composed of an ensemble of 24 statistically independent MD simulations (obtained again by randomizing velocities), initiated from the Stage 2 trajectory in which lipid advanced from the IC side to the central region of the groove (indicated by the red arrow in Supplementary Figure 13a).

Stage 1 of this multi-scale iterative protocol was run with NAMD 2.10 as described above. Stages 2 and 3 were carried out with the latest version of the ACEMD software[57] and run according to the protocol developed at Acellera and implemented by us previously[37,42,57,58]. The conditions include the PME method for electrostatic calculations, 4 fs integration time-step, and the standard mass repartitioning procedure for hydrogen atoms. The computations were conducted under the NVT ensemble (at $T = 298$ K), using the Langevin Thermostat with the Langevin Damping Factor set to 0.1.

All the simulations implemented the CHARMM36 force-field parameters for proteins[59], lipids[60,61], and ions[62].

**Dimensionality reduction using tICA.** To facilitate analysis of lipid translocation process in the ensemble MD simulations of the WT nhTMEM16 embedded in POPC membranes, we performed dimensionality reduction using the tICA approach, the time-structure based independent component analysis, as described

before[37]. To define the tICA space we used the following eight dynamic variables extracted from the analysis of the ensemble MD trajectories: (1)-the z-directional distance between the phosphorus atom of the translocated lipid and the $C_\alpha$ atom of R432; (2)-the minimum distance between T333 and Y439; (3–5)-the minimum distances between Y439 and R432; between E313 and R432; and between E318 and R432 (defined as distance between carbonyl oxygen of Glu and sidechain nitrogen of Arg); (6–8)-the distances between the phosphorus atom of the translocated lipid and residues E313, E318, and R432. For additional details, see Results and Supplementary Figure 14, 15.

**Definition of the hydrophilic groove and its compartments**. To quantify hydration of the groove and of its various compartments and to facilitate description of lipid dynamics along the groove we defined several volumes and counted number of waters and lipids in these volumes. Thus, a water molecule was considered to belong to the groove (see Supplementary Figure 5) if any of its atoms was within 3 Å of the sidechains of the following residues lining the groove area: 302, 306, 310, 313, 333, 336, 337, 340, 341, 344, 348, 352, 367, 370, 371, 374, 377, 378, 381, 382, 385, 432, 436, 439, 440, 444, 447, 451, 455, 499, 501, 505, 509, and 513. Similarly, a lipid was deemed to be in the groove if its phosphorus atom was within 5 Å of the sidechains of the same set of residues. In addition, water molecule (or lipid) was considered to belong to the extracellular vestibule of the groove if z-coordinate (along membrane normal) of any atom of the water molecule (or the phosphorus atom of the lipid) was found between the x/y (membrane) planes defined by the $C_\alpha$ atoms of residues T381 and Q436 (see Fig. 4b).

**Data availability**. Data supporting the findings of this manuscript are available from the corresponding authors upon reasonable request.

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

## Acknowledgements

The authors thank members of the Accardi and Weinstein labs for helpful discussions. This work was supported by NIH Grant R01GM106717 (to A.K.M. and A.A.), an Irma T. Hirschl/Monique Weill-Caulier Scholar Award (to A.A.), by the Basic Science Research Program through the National Research Foundation of Korea (N.R.F.) funded by the Ministry of Education, Science and Technology (grant 2013R1A6A3A03064407 to B.-C. L.), the 1923 Fund and Cofrin Center for Biological Information (to H.W. and G.K.). The computational work used the Extreme Science and Engineering Discovery Environment (XSEDE, account TG- MCB120008), which is supported by National Science Foundation grant number ACI-1053575, the Anton 2 computer at the Pittsburgh Supercomputing Center (PSC) through Grant *R01GM116961* from the National Institutes of Health (The Anton 2 machine at PSC was generously made available by D.E. Shaw Research), resources of the Oak Ridge Leadership Computing Facility (ALCC allocation BIP109 and Director's Discretionary allocation) at the Oak Ridge National Laboratory, which is supported by the Office of Science of the U.S. Department of Energy under contract no. DE-AC05-00OR22725, the computational resources of the David A. Cofrin Center for Biomedical Information in the HRH Prince Alwaleed Bin Talal Bin Abdulaziz Alsaud Institute for Computational Biomedicine at Weill Cornell Medical College.

## Author contributions

B.-C.L., G.K., H.W. A.K.M., and A.A. designed the experiments, B.-C.L. and M.F. performed experiments, G.K. performed the molecular dynamics simulations, B.-C.L., G.K., H.W., and A.A. analyzed the data and wrote the paper. All authors edited the manuscript.

## Additional information

**Competing interests:** The authors declare no competing interests.

