## [Peer Review File · Nature Communications]

Reviewers' comments:

Reviewer #1 (Remarks to the Author):

In their manuscript, the authors describe results from an interdisciplinary study that combines lipid and ion transport assays with molecular dynamics simulations to provide novel insight into the mechanism of transmembrane lipid and ion permeation catalyzed by the scramblase nhTMEM16. By studying the transport behavior of different purified and reconstituted nhTMEM16 proteins containing mutations in the putative lipid translocation path, the authors identify a region where mutations had a large and predominantly inhibitory effect on lipid transport. This region is located at the extracellular part of a hydrophilic and membrane-exposed furrow and extends towards the center of the lipid bilayer. By employing detailed and extended molecular dynamics simulations, they find that the same region is poorly populated by lipids, in line with the proposal that the diffusion of lipids across this region would be a rate-limiting step for translocation. In their simulations the authors also identify conformational changes leading to an altered pattern of interactions of charged residues in the same region, which are correlated with lipid movement. A subsequent mutagenesis study of the residues identified in the simulations revealed the conservation of the charge of the respective residues to be important for transport activity. In summary, I think that this is a very strong paper that combines wealth of experimental data of exceptional quality with carefully performed simulations to address a novel transport mechanism that is to date still poorly understood. The transport data are conclusive and provide the first in-vitro mutational study of a lipid scramblase. Whereas computer simulations are to some degree speculative by nature, the correlation between experiments and simulations is intriguing. In combination, the described results provide an interesting mechanism that is in general agreement with previous studies but also contains multiple novel features. I thus think that the manuscript is a strong candidate for publication in Nature Communications. I have only few remarks, which are listed below.

Specific remarks:

- The described conformational changes in the lipid translocation path are interesting, but since they appear to be irreversible at the timescale of the simulation, and have never been observed experimentally, they, deserve a better documentation. For that purpose a stereo figure of a stick representation of the conformation shown in Fig. 1g (red) in comparison to the X-ray structure could be shown. In any case, the claim that the structure would require some rearrangement for lipid conduction is plausible.
- There is no evidence for a strong salt-bridge between E313 and R432 in the X-ray structure (as mentioned in line 242). Whereas the density of R432 is well-defined in the electron density, E313 appears to be more mobile. Nevertheless, due to their proximity some long-range coulombic interactions between both residues are likely.
- The detailed lipid-protein interactions observed in Fig. 6 a-d are difficult to grasp. An improved representation would be of help.
- A schematic figure summarizing the described mechanism at the end of the manuscript might facilitate to convey the main finding of the study.
- In their comparison with TMEM16A, the authors might want to refer to two recently published studies that describe the structure at higher resolution.
- In the scrambling data of mutants shown in Fig. 7a-f, the slower kinetics is visible in R432A and R432L, whereas the fluorescence decay is fast for E313A and E313Q but appears to level off at a high level. Is there a mechanistic explanation for this different behavior?

Minor:

I could not identify the circle denoted by an asterisk in Fig. 5.

Reviewer #2 (Remarks to the Author):

This very nice manuscript from the Accardi lab uses a combination of computational simulations and experimental mutagenesis to define the conduction pathway for lipid through the lipid scramblase nhTMEM16. Lipid scramblases are thought to play important roles in cell signaling, but yet the molecular mechanisms of lipid scrambling was largely inaccessible to study until the Accardi lab purified and reconstituted the first TMEM16 scramblase in 2013. Subsequently, crystallization of a closely related TMEM16 scramblase by the Dutzler lab in 2014 provided important clues into how phospholipids move through the protein. The TMEM16 crystal structure suggested that lipids move through a groove on the surface of the protein facing the lipid bilayer. Recent molecular dynamics simulations accompanied by a few mutagenesis experiments support this hypothesis, but this manuscript is the first to systematically explore the lipid permeation pathway using a combination of molecular dynamics simulations and experimental mutagenesis. The paper is clearly presented and the wealth of data presented provide very important understandings of the process of lipid transport by the TMEM16 proteins. The data and conclusions will add significantly to thinking in the field. The paper should be of broad interest to investigators in the fields of ion channels, lipid transport and metabolism, membrane protein structure-function, and cell signaling.

I have only relatively minor comments.

Line 82 and elsewhere – the authors should specify nhTMEM16 rather than using the generic term TMEM16.

Line 133. “placed” seems an odd word.

Line 173. References 13,15,16 should probably include reference 14.

Line 175-186. I found this explanation hard to follow. In particular, the statement on line 180 that “the experiments overestimate the activity of the channels” was puzzling because I was unsure exactly that “activity” meant (activity = A ? or single channel open probability?). Also, if one active protein per vesicle can dissipate the ion gradient but the vesicles actually contain 5-6 active proteins, it seems the experiments UNDER-estimate the fraction of active proteins, which seems to be what the next sentence “Thus, we...” is saying.

Line 256. Can the authors be more quantitative about the frequency of occurrence of the “rare mode” relative to the common mode?

Line 698. Phosphate atom > phosphorous atom/phosphate group?

Figure 3. What is the black starburst labelled “D” in panel a? Also, the arrow from Ca makes it look like Ca is closing the gate.

Figure 4. Could the authors clarify the definition that a lipid headgroup was considered to be in contact with a residue if its phosphate was within 7\AA of any atom of the residue? Is center-center distance = 7\AA or = VDW radii + 7\AA ? If the latter, please justify.

Reviewer #3 (Remarks to the Author):

The manuscript reports on a combined experimental and computational study focusing on some of the mechanistic and structural aspects of lipid scrambling in nhTMEM16, a fungal scramblase and a nonselective ion channel of the TMEM16 superfamily. From a number of recent structural, biochemical, and simulation studies, it has been fairly well established that a membrane-exposed hydrophilic groove on the surface of nhTMEM16 serves as the phospholipid translocation pathway. The same pathway may potentially also serve as the ion permeation pathway in the ion-conducting members of the family. Similar findings and conclusions regarding the involvement of the

hydrophilic pathway in phospholipid translocation are also reported in the present study. Lipids are observed to populate the pathway and use it to move from one leaflet to the other during the performed simulations, and on the experimental side, their extensive mutagenesis data strongly support the involvement of pathway-lining residues in the process, as their mutations results in modulation of the rate of the scramblase activity to different degrees. The additional insight provided by the present study is a more in-depth examination of the lipid translocation pathway on the extracellular side, including a "gating" mechanism materialized by a salt bridge on the extracellular entry point into the translocation pathway, as also highlighted in the title of the manuscript.

The manuscript includes large data sets, both on the experimental and on the computational sides. This is commendable. The computational part of the study focuses on describing lipid-protein interactions and their effect on the extracellular entry of the lipid into the pathway concluding that rearrangement of a triad network of charged residues is necessary to allow the opening of the extracellular entrance into the central constriction of the pathway (the gate). On the experimental side, extensive mutagenesis has been performed, focusing on the extracellular half of the lipid translocation pathway and providing additional molecular details. However, it is difficult to bring the simulation and experimental parts of the study together in most places, especially with regard to the salt-bridge gate, which is represented as the mainstay of the study, and for which the experimental part does not seem to provide any support. The disconnect between the two parts of the study also manifests itself in the conspicuous absence of scrambling activity data from the simulations (number of phospholipids crossing), whose trend should have been compared to the experiments for the wildtype and major mutants. Therefore, some key experiments and simulations, which could have strengthened the connection between the two data sets are currently missing, and some of the results are not quite in line with the authors' interpretation and the presented model. I would like very much to see their data published, but this disconnect presents a major issue that needs to be satisfactorily addressed and clarified for a joint experimental/computational paper.

The following provides a more detailed description of the issues that need to be addressed and clarified. The first few points go to the heart of the main theme of the study and the currently lacking convergence of the experimental and simulation data. Either the connection between the experimental and simulation results needs to be strengthened, e.g., by some additional experiments and simulations or additional clarification, OR the interpretation and presentation of the results should be done in a majorly restructured manner to avoid confusion. A related issue which is somewhat the down side of inclusion of such a large data set is the difficulty of unifying all the data into a working model. I will refrain from listing all such cases. Instead, I would recommend addressing some of the major deviations in the text explicitly, so the reader is aware of them.

While the experiments concentrate on reporting the rate of scrambling as the main readout, there is essentially no report on the number of translocating lipids in the simulations and how the number was affected by the mutations, which are shown to reduce the rate by orders of magnitude experimentally. This is conspicuously missing. I do not expect to get a perfect match between the experiment and simulations, or even the order of magnitude agreement in the absolute values, but at least the trends need to be reproduced, i.e., one needs to see a reduction in the number of flipped lipids, or reduction in their diffusion along the track, or some other parameters quantifying the movement of lipids, due to the mutations, in order to be able to reconcile the two approaches. Instead of making this direct comparison, the authors examine and discuss lipid occupancy of the pathway, which is not only a good measure, but also not showing much difference between the wildtype and the simulated mutants. How are the simulation results substantiating the observed experimental changes in lipid translocation rate? This is one of the clear disconnects that can be avoided by a more direct comparison.

As a more specific example of the point above, how many lipid scrambling events (full

translocation from one leaflet to another) are captured in the A385W simulation? If only a single event was captured, please comment on the statistical certainty of your result. Do you see any lipid scrambling in the WT simulations? Why lipid scrambling is captured in A385W which is supposed to reduce scrambling by >100 fold as shown in the mutagenesis results? Throughout the manuscript, the simulation data from WT constructs are comparable to the two mutants, are there any features that only exist in WT simulations or the mutated ones? In other words, are there any details revealed by the simulations that could help substantiate the functional differences between the mutants and the WT?

The extracellular salt bridge is described as a gating mechanism. There are several issues with this aspect, which seems to constitute the main finding of the manuscript (highlighted in the title). (1) First of all, structural breathing of this salt bridge may simply represent one of many fluctuating features within the pore that are coupled to the translocation of the substrate. Though interesting, it is not necessarily a "gate" that "regulates" the lipid translocation, since its motion does not seem to be coupled/controlled by anything else. There could be other motions in the protein lumen that are needed to allow lipids to pass. (2) This becomes more of an issue, since the simulations do not provide a convincing mechanism as to how the breakage of the salt bridge is coupled to increased lipid traffic (again, lack of data quantifying lipid movement). Rather, the authors calculate lipid coordination of the residues as a measure of "openness" of the gate. How can this indicate the increased rate of lipid translocation? This could be simply related to the higher exposure of the residues to interact with lipids. (3) Most importantly, there is no experimental data in the paper supporting the role of the salt bridge in gating/closing the pathway. The experiments in which this salt bridge has been manipulated/abrogated all result in a less effective translocation pathway (the opposite effect), rather than providing evidence supporting a more open path. This is a major problem here, and I suggest that this point (gating/breathing motion of the salt bridge) be substantially deemphasized and reduced to perhaps an interesting observation within the simulations, rather than the mainstay of the paper.

If the gating mechanism is truly effective, one additional simulation, in which this can be further nailed down (at least within the simulation domain of the study), would be to keep the gate "open" through some restraints and show increased lipid movement (flipping, or at least some diffusive indicators along the pathway or any quantitative measure pointing in the direction of more efficient motion). Such an additional simulation can at least provide evidence on the computational side that the formation of the salt bridge can hamper permeation. Still, in the absence of experimental support for the effect, I would largely reduce the tone on the importance of this feature.

In the A385W simulation, it is stated that the bulky side chain of Trp induces an open conformation of the groove, but in the Q436W simulation, introducing W at Q436 locks the groove in a closed conformation. It is expected that authors provide clarification as to how Trp brings about opposite effects at the two sites in the simulation (perhaps describing/showing the mutation sites in more detail). Unfortunately, both mutants have similar phenotypes in the experiments, namely greatly reducing the scrambling activity (by >100 fold), which would be hard to reconcile with the observed opposite effect of W at these two different sites.

The authors did extensive W-scanning mutations on residues lining the upper half of the groove; 17 residues have been tested in total. However, not every single residue from the upper groove has been tested. For example, N435 is one of the residues that form the strongest interactions with lipids (Fig. 4). However, this residue is not tested by mutagenesis, while the other weaker interactors were tested. I am fully cognizant of the potential technical difficulties here, and the fact that the authors are already providing a large data set, but considering this mutant would strengthen the connection between the simulations and the experiment.

Similarly, all the 17 mutated residues are from the extracellular half of the groove even though the authors mention that certain residues from the cytoplasmic half, for example R505, also form strong interactions with lipids inside the groove. This is somewhat surprising. If the authors would

like to remain focused only on the extracellular half, in which case it may be helpful to provide a heads up to the reader, so they are not surprised by the lack of any data for the cytoplasmic half.

Previous studies suggest that phospholipid scrambling does not require any ionic electrochemical driving force, however, the nonselective ion transport by scramblases relies on scrambling and a leak current is associated with lipid transport. In this manuscript, authors found that the charge reverse mutation R432E retains WT-like channel activity while not conducting lipids. The interpretation of this result should be discussed.

Based on Fig. 2g-2h, and Fig. 3b, it is stated that the mutations that affect ion channel activity mostly overlap with those that influence lipid scrambling, with the sole exception of N317. The bar plots in Fig. 2g-2h, and Fig. 3b are too compressed and difficult to read. Please separate the bars between each mutant. It's hard to tell any correlation between Fig. 2g-2h and Fig. 3b merely by looking at the figures (for example, S382W severely affects scrambling while does not affect ion channel activity at a similar extent). Analysis of correlation would be helpful to interpret the data. N317W strongly reduces AN317W(0 Ca²⁺) by ~7-fold, while has not much influence on AN317W(+Ca²⁺). Is there a way to reconcile the discrepancy? If not, perhaps just mention it explicitly. I understand that there is a large data set here (and that is great) and not everything can be perfectly fitting, but if there are clear deviations the reader should be alerted.

The salt bridge rearrangement involves the breaking of R432-E313 and the bonding of R432-E318. In Suppl. Fig. 7d upper panel, the red trace almost overlaps with the black one, does that mean the R432 is forming bond with E313 and E318 at the same time? What's the behavior of the lipids in this situation?

Could the dilation of the central constriction in the A385W simulation be mainly due to the rotation of Tyr439 (Fig. 6g), rather than the movement of TM3 and TM6. One has to better quantify the separation of the two helices in order to associate the dilation to helix separation. From SFig. 7, T333-Y439 separation happens in several systems. Do they see similar sequential events of R432 sidechain rearrangement and Y439 rotation as described in the A385W simulation?

Minor points:

It is stated that the E313-R432 salt bridge in the crystal structure serves as a gate. If I am not mistaken, E313 and R432 do not form ionic bond in the two available nhTMEM16 structures 4WIT.pdb and 4WIS.pdb (authors use the latter for their simulations).

Lines 151-153: "...TM4 residues (N317, F330, T333 and V337) have low to no impact on scrambling..." N317 is actually on TM3 rather than TM4. A recent study on nhTMEM16 shows that T333 is functionally important and that mutating this Thr to the corresponding amino acid Val in the TMEM16A channel homolog results in very little scrambling. A brief mention/discussion would be in order.

Line 288-290: Fig. 6i should be Fig. 6h. What distance is measured between Y439 and lipid? The distance between Y439 and the lipid decreases and then increases as the lipid bypasses Y439, but there is no data showing that Y439 coordinates the headgroup during the lipid translocation. As shown in Fig. 6g, Y439 side chain rotates away toward the membrane rather than coordinating the lipid. Please clarify.

What is the average number of lipids in each groove of the individual construct? From SFig. 4a and 4e, A385W has more lipid occupancy than WT1 in PROA, and Q436W has more lipid occupancy than both WT1 and WT2 in PROB. Please explain why these mutants have even better lipid occupancy compared to the WT constructs.

Fig. 7: The R432L mutation is not mentioned in the text.

Lines 68-71: Figure 1 only shows TM3-6, TM7 is missing. Isn't it better to also show TM7 along with the other 4 helices in Panel B of Figure 1? It is introduced as one of the cavity-lining helices. Which pairs of amino acid were chosen to approximate the width of the groove and why?

Fig. 2b: No fluorescently labeled (green dots) or bleached (black dots) lipids are shown on the vesicles. Difficult to compare with Fig. 2a.

The numbering of the panels in SFig. 4 is not matching with the description in the text.

In SFig. 8, the upper panel is PROB and lower panel is PROA, while in SFig. 7, the panels are not labeled. I assume the top panels are PROA. Please label the panels and make them consistent. The scaling should also be consistent in SFig. 7.

Lines 255-257: "... (Fig. 5, blue circle denoted by "*")..... a rare mode of lipid penetration ... was observed." There is no blue circle denoted by * in Fig. 5. In which monomer of A385W does the rare mode take place? And the time frame for this rare mode?

The numbering of the panels in SFig. 7 is not matching the description in the text.

There is a mention of Fig MD1 in the text. Not clear what it refers to.

Figure 1. Description of the dashed (solid) lines in panel B is confusing. Fix.

We thank the reviewers for their appreciation of our manuscript and their constructive suggestions. We carried out the additional experiments and computations suggested in the critiques and these, we believe, have strengthened our conclusions and improved the manuscript. Our responses to the Reviewers' comments and suggestions are detailed below. All changes to the text have been highlighted in blue to indicate where the manuscript has been modified. With these changes we hope that the manuscript can now be accepted in *Nature Communications*.

We have taken the following steps to address the main concerns raised by the reviewers:

A1) We carried out and analyzed an additional large set of new MD simulations of WT nhTMEM16. These simulations show multiple events of lipid transfer from the inner to the outer leaflet. We find that the gating steps that allow the lipid to translocate through the pathway in these new simulations recapitulate, in the expected reverse order, the ones we described in the original version of the manuscript for lipid transfer from the extracellular (EC) to the intracellular (IC) leaflets.

A2) As requested, we simulated the scrambling-impaired E313A/E318A/R432A triple mutant. The results match and explain the experimental phenotype (see below response to Reviewer 3, Figure 1). We prefer not to include these results in the revised manuscript because the text is already very dense and, in our opinion, these simulations are confirmatory and do not add novel insights.

A3) We improved presentation of several figures as per the reviewers' requests.

Detailed responses to the Reviewers' comments and notation of the places in the text where particular issues have been addressed and additions or modifications were made.

Reviewer #1

"...I thus think that the manuscript is a strong candidate for publication in Nature Communications. I have only few remarks, which are listed below."

We thank the reviewer for the appreciative comments on our manuscript.

Specific remarks:

1) The described conformational changes in the lipid translocation path are interesting, but since they appear to be irreversible at the timescale of the simulation, and have never been observed experimentally, they, deserve a better documentation. For that purpose a stereo figure of a stick representation of the conformation shown in Fig. 1g (red) in comparison to the X-ray structure could be shown. In any case, the claim that the structure would require some rearrangement for lipid conduction is plausible.

We thank the reviewer for this excellent suggestion. We added a figure in the discussion (new Figure 10, discussed in page 17 of the revised manuscript) which illustrates the steps of our proposed gating mechanism and the conformational rearrangements undergone by the groove region.

2) There is no evidence for a strong salt-bridge between E313 and R432 in the X-ray structure (as mentioned in line 242). Whereas the density of R432 is well-defined in the electron density,

E313 appears to be more mobile. Nevertheless, due to their proximity some long-range coulombic interactions between both residues are likely.

Thank you for pointing this out. We modified language throughout the manuscript to remove reference to “salt-bridge” and replace it with either “electrostatic interaction” or simply “interaction”.

3) The detailed lipid-protein interactions observed in Fig. 6 a-d are difficult to grasp. An improved representation would be of help.

We hope that together with new structural representations that we added to the revised text (Figure 7, Supplementary Figure 13; see response to the next point below) the figures now better convey the central message of the paper related to extracellular gating mechanism. It was not our intention to provide detailed description of lipid-protein interactions in Fig. 6a-d. Rather, the structural models in Fig. 6a-d were conceived primarily to show re-arrangements in the E313/E318/R432 triad and in the T333-Y439 pair of residues, which we believe they do capture. In addition, aligned with the text, our goal was to show in these panels accumulation of lipids at the EC gates. We believe that the figure shows how lipids can arrange either at the side entrance to the groove (purple lipid in panels a-c) or insert into the EC vestibule of the groove (green and blue lipids in panels b-c).

4) A schematic figure summarizing the described mechanism at the end of the manuscript might facilitate to convey the main finding of the study.

We appreciate the suggestion and have added a new figure in the discussion where we illustrate the key conformational steps undergone by the pathway (new Fig. 10). Additionally, we also illustrate the key steps underlying our proposed mechanism in the new Fig. 7a.

5) In their comparison with TMEM16A, the authors might want to refer to two recently published studies that describe the structure at higher resolution.

We have added the new references. Thank you.

6) In the scrambling data of mutants shown in Fig. 7a-f, the slower kinetics is visible in R432A and R432L, whereas the fluorescence decay is fast for E313A and E313Q but appears to level off at a high level. Is there a mechanistic explanation for this different behavior?

The reviewer raises an excellent point that is addressed in the revised manuscript (page 6). The level at which the fluorescence plateaus measures the fraction of liposomes that do not contain an active protein and/or are inaccessible to dithionite, f_0 . Thus, the simplest interpretation of a reduction in the plateau level is that the fraction of liposomes with at least one active protein has decreased. We find that there is some amount of variability in the plateau level between different liposome preparations, even for WT preps. For this reason, every mutant is compared to a WT reconstitution done side-by-side. In our analysis we assume that f_0 of the mutants is the same as that of the WT from the same reconstitution, an assumption supported by our finding that the reconstitution efficiency of the mutants is comparable to that of the WT (Supplementary Fig. 2). Thus, a change in the plateau level fluorescence is reflected in a reduced ‘macroscopic’ scrambling rate. It is important to note here that the scrambling rates derived from our analysis are macroscopic parameters that include the number of molecules per vesicle, their “open probability” and their “conductance”. Thus, it is possible that some of these mutations might

favor a long-lived inactivated state of the scramblase, without affecting their conductance. This is discussed in Page 6.

7) I could not identify the circle denoted by an asterisk in Fig. 5.

We apologize for this omission. The asterisk is now shown in Fig. 5.

Reviewer #2

8) The paper should be of broad interest to investigators in the fields of ion channels, lipid transport and metabolism, membrane protein structure-function, and cell signaling.

We thank the reviewer for the overall positive feedback on the manuscript.

9) Line 82 and elsewhere – the authors should specify nhTMEM16 rather than using the generic term TMEM16.

Thank you. We specified nhTMEM16 where appropriate.

10) Line 133. “placed” seems an odd word.

The word “placed” has been replaced by “determined”.

11) Line 173. References 13,15,16 should probably include reference 14.

Thank you, we added the missing reference.

12) Line 175-186. I found this explanation hard to follow. In particular, the statement on line 180 that “the experiments overestimate the activity of the channels” was puzzling because I was unsure exactly that “activity” meant (activity = A? or single channel open probability?). Also, if one active protein per vesicle can dissipate the ion gradient but the vesicles actually contain 5-6 active proteins, it seems the experiments UNDER-estimate the fraction of active proteins, which seems to be what the next sentence “Thus, we...” is saying.

We apologize for the confusion. What we meant by ‘activity’ is a combination of open probability and conductance. We reworded this to ‘ion throughput’.

The assay overestimates the ion throughput of the channel, as channels with low throughput (low conductance and/or P_o) will be as effective in emptying vesicles as those with high throughput (high conductance and/or P_o). For example, a high activity channel with a high conductance of 10^8 ion s^{-1} and a $P_o=50\%$ will dissipate the content of a liposome ($\sim 10^6$ ions) in <0.01 s. For this reason, the effects of the mutations are underestimated, for example a 10-fold reduction in the throughput of the above-mentioned channel would still result in an ‘active’ channel.

To clarify this issue we reworded the paragraph on page 8 as follows:

“Thus, we can estimate that a significant reduction (>2 -fold) in the fraction of liposomes containing at least one active channel corresponds to a >100 -fold reduction in the ion throughput of the channel. This implies that the assay underestimates the effects of the mutations.”

13) Line 256. Can the authors be more quantitative about the frequency of occurrence of the “rare mode” relative to the common mode?

We addressed this in the revision. Briefly, the rare mode of lipid tail insertion was observed in 2/8 protomers of nhTMEM16 in the original set of simulations (one protomer of the A385W construct, and one of the WT1 system). Notably, the same rare mode was also observed in the new set of simulations we added in the revision (see section A(1), at the beginning of this response document) and it is discussed in the context of Supplementary Figure 10. We also mention in the manuscript that this mode of blocking the lipid transfer path was observed, also as a rare event, in the computational analysis of lipid scrambling by the G protein-coupled receptor opsin reported recently (Morra et al., 2018).

14) Line 698. Phosphate atom > phosphorous atom/phosphate group?

We renamed phosphate atom to phosphorous atom in the revised manuscript.

15) Figure 3. What is the black starburst labelled “D” in panel a? Also, the arrow from Ca makes it look like Ca is closing the gate.

The “D” indicates the addition of detergent at the end of the experiment to solubilize the liposomes to determine the total trapped volume.

We moved the arrow to emphasize that Ca²⁺ opens the channel.

16) Figure 4. Could the authors clarify the definition that a lipid headgroup was considered to be in contact with a residue if its phosphate was within 7Å of any atom of the residue? Is center-center distance = 7Å or = VDW radii + 7Å? If the latter, please justify.

Our definition is for center-to-center distance. This has been clarified in the Figure 4 captions.

Reviewer #3

We thank the Reviewer for his/her feedback and for thoughtful comments. We believe that by inclusion of the responses to these comments the manuscript improved significantly. As stated at the beginning of the response document we have taken several steps to address the main concerns raised by the reviewer.

One of the reviewer’s main concerns was that in our original set of simulations we observe only a single scrambling event. To address this, we carried out additional simulations using an enhanced sampling method that allowed us to capture several full lipid translocation events. These simulations use an iterative protocol described most recently in the study of opsin-mediated lipid translocation (Morra et al., Structure, 2018). In this approach, simulations are run in multiple consecutive stages (as described in the revised manuscript), so that the initiation of each stage is informed by the output from the previous stage. Each stage was composed of multiple replicate simulations for a total of 24 statistically independent runs (500-600 ns each) for the wild type nhTMEM16. In this set, a lipid transferred completely from the IC to EC leaflet in 7/24 simulations. Most importantly, we find that the gating mechanism that opens the lipid translocation pathway is the same as that we described in the original manuscript for lipid transfer in the A385W construct from the EC to the IC side (Fig. 6). Briefly, when a lipid approaches the groove from the IC side, opening of the T333-Y439 constriction enables its translocation to the EC side ending in a complete flip. For the flipped lipid to merge with the EC leaflet, the E313/E318/R432 gate opens following the sequential disengagement of R432 from E313 and E318. Please see the revised manuscript and Figs. 7 and S13-S15 for more details of these new results.

In the following we address the Reviewer’s comments point-by-point.

17) However, it is difficult to bring the simulation and experimental parts of the study together in most places, especially with regard to the salt-bridge gate, which is represented as the mainstay of the study, and for which the experimental part does not seem to provide any support.

We agree with the Reviewer that the gating activity resulting from the dynamic rearrangement of the network of interactions among the charged residues is indeed a main finding of the study supported by a large number of separate and mutually reinforcing findings from the analysis of the trajectories. We respectfully disagree with the reviewer's view that *the experimental part does not seem to provide any support*. Rather, our experimental and computational data present a cohesive picture supporting our proposed gating mechanism. We clarified the direct lines of evidence where the experimental and computational data mutually support each other, and now explicitly discuss the points where open questions remain.

Here we briefly summarize what we believe are the key mutually supportive pieces of evidence:

- i. We show that the only substitutions tolerated at positions E313 and R432 are charge conserving mutations (E313D and R432K) (Fig. 8). This indicates that these residues are essential for scrambling.
- ii. Our simulations identify three additional residues – E318, Q436, and Y439 – that rearrange during scrambling. Our experiments support this notion, as all 3 positions are important for scrambling mediated by nhTMEM16 (Fig. 2, 9).
- iii. Our simulations suggest that E313, E318 and R432 form a dynamically rearranging network of residues. Our combined mutations support this idea: disruption of individual or combined parts of the network results in profoundly impaired scrambling (Fig. 9). If the interactions between these residues were pairwise and static, then we would expect that their combined mutations should favor opening of the pathway. In contrast, we find that this is not the case (Fig. 9), consistent with the idea that scrambling entails a dynamic rearrangement of the organization of these residues.

Thus, our data supports our hypothesis that the amino acids located at the extracellular entryway of the lipid groove form a dynamically rearranging rather than static network of interacting residues.

18) “The disconnect between the two parts of the study also manifests itself in the conspicuous absence of scrambling activity data from the simulations (number of phospholipids crossing), whose trend should have been compared to the experiments for the wildtype and major mutants.”

“While the experiments concentrate on reporting the rate of scrambling as the main readout, there is essentially no report on the number of translocating lipids in the simulations and how the number was affected by the mutations, which are shown to reduce the rate by orders of magnitude experimentally. This is conspicuously missing.”

Our experimental data suggests that scrambling occurs at a rate of $\sim 10^4$ lipid s^{-1} , which corresponds to an average lipid permeation time of ~ 100 μs . Thus, it is not surprising that in the ~ 11 μs long MD trajectories originally reported we observed a single transition event. We performed additional simulations using an enhanced sampling algorithm, see above, to increase the sampling rate of the permeation events. We generated 24 additional statistically independent MD trajectories, and observed lipid scrambling events in 7 of these. These simulations serve to strengthen our mechanistic conclusions about the involvement of the EC gates in lipid scrambling (see revised manuscript and also detailed responses below). We stress, however, that the data obtained from this enhanced sampling algorithm cannot and should not be used in quantitative comparisons of the scrambling rates in experiments and

simulations, or between wild type and mutants, as the extent of sampling for the different constructs is not the same.

19) I do not expect to get a perfect match between the experiment and simulations, or even the order of magnitude agreement in the absolute values, but at least the trends need to be reproduced, i.e., one needs to see a reduction in the number of flipped lipids, or reduction in their diffusion along the track, or some other parameters quantifying the movement of lipids, due to the mutations, in order to be able to reconcile the two approaches. Instead of making this direct comparison, the authors examine and discuss lipid occupancy of the pathway, which is not only a good measure, but also not showing much difference between the wildtype and the simulated mutants. How are the simulation results substantiating the observed experimental changes in lipid translocation rate? This is one of the clear disconnects that can be avoided by a more direct comparison.

We thank the reviewer for this set of comments that prompted us to introduce new data and specific arguments. As we discuss above, in response to point #17, we show how predictions from our simulations are met by our experimental results and provide an internally mechanistic framework to interpret our scrambling data. The reviewer asks for a direct comparison of the experimentally determined rates of lipid transfer to those seen in the simulations. However, as we discuss above (see response to point #18), lipid scrambling occurs on a relatively slow time scale (a lipid crosses the membrane in ~100 μ s). Thus, to provide an accurate and statistically significant comparison between the rates in experiments and simulations several milliseconds worth of simulations for the wild type and mutant proteins are needed. To address this concern, we performed a new set of large-scale unbiased MD simulations where we capture multiple events of lipid scrambling from the IC to the EC leaflet in the wild type system. Importantly, using the same approach we show why the triple mutant E313A/E318A/R432A mutant does not scramble lipids (see below, response to point #21), further reinforcing the harmony of the experimental and computational results.

“As a more specific example of the point above, how many lipid scrambling events (full translocation from one leaflet to another) are captured in the A385W simulation? If only a single event was captured, please comment on the statistical certainty of your result. Do you see any lipid scrambling in the WT simulations? Why lipid scrambling is captured in A385W which is supposed to reduce scrambling by >100 fold as shown in the mutagenesis results? Throughout the manuscript, the simulation data from WT constructs are comparable to the two mutants, are there any features that only exist in WT simulations or the mutated ones? In other words, are there any details revealed by the simulations that could help substantiate the functional differences between the mutants and the WT? “

As we describe in our manuscript (Page 10), in the original set of WT simulations we observe several partial lipid penetration events in the groove and a full “flip” in one protomer of the A385W mutant. The partial events recapitulated well what we saw in the full lipid translocation, supporting our analysis of the mutant’s results. We have since performed additional simulations (see point A1 above) using an enhanced sampling approach where we capture multiple lipid translocation events that recapitulate the mechanistic steps we identified in our original set of simulations. These observations support our proposed mechanism.

Our simulations suggest a mechanism of action for the A385W and Q436W (as we described in the original manuscript on Page 14): the pathway is non-conductive because of the stabilizing interactions of W385 or W436 with E313. This interaction is not present in the wild type protein and adds to the energy cost of opening the lipid translocation pathway in the mutants. For the Q436W system, the H-bond keeps positions 313 and 436 in close proximity for

the duration of the simulation in both protomers (Figure S7D of the revised manuscript), hence the effect on scrambling. For A385W, on the simulation time-scales the H-bond between W385 and E313 is formed in one of the protomers (Figure S16 of the revised manuscript) which is blocked and does not transfer lipid. In the other protomer, the H-bond is not stable and scrambling occurs.

20) The extracellular salt bridge is described as a gating mechanism. There are several issues with this aspect, which seems to constitute the main finding of the manuscript (highlighted in the title).

(1) First of all, structural breathing of this salt bridge may simply represent one of many fluctuating features within the pore that are coupled to the translocation of the substrate. Though interesting, it is not necessarily a “gate” that “regulates” the lipid translocation, since its motion does not seem to be coupled/controlled by anything else. There could be other motions in the protein lumen that are needed to allow lipids to pass.

We respectfully disagree with the reviewer’s view that the salt bridge is not a gate for several reasons.

- 1) A ‘gate’ is an energy barrier that (i) prevents substrate movement and (ii) whose height is dynamically regulated by a rearrangement in the protein (i.e. can open and close). Our data supports the notion that a rearrangement of the E313-R432 pair is necessary for substrate translocation (Fig. 2, 8, 9, see response to point #17 above).
- 2) We describe in the manuscript (Page 10) how the disengagement of the E313-R432 pair gate is regulated by specific interactions with lipids. Further, our analysis (Fig. 6, 7; Suppl. Fig. 15) shows that the rearrangements we describe are not random, uncorrelated movements in the protein, but rather that they form a highly concerted set of conformational changes requiring precise interactions between residues and lipids in the pathway. Disruption of any of these interactions has catastrophic consequences on scrambling (Fig. 2, 8, 9, response to point #20 below).

We agree with the reviewer that rearrangement of the salt bridge is likely coupled to “*other motions in the protein lumen*”. We discuss this in detail in the manuscript in the context of the gating with which it is connected.

(2) This becomes more of an issue, since the simulations do not provide a convincing mechanism as to how the breakage of the salt bridge is coupled to increased lipid traffic (again, lack of data quantifying lipid movement). Rather, the authors calculate lipid coordination of the residues as a measure of “openness” of the gate. How can this indicate the increased rate of lipid translocation? This could be simply related to the higher exposure of the residues to interact with lipids.

To describe openness of the gate we use several measures, not just “lipid coordination of the residues as a measure of “openness” of the gate”. In addition to lipid coordination, we report distances between three residue pairs (313-432, 318-432, 333-439, Figs. 6-7, Supplementary Fig. 15), as well as water counts in the different compartments of the groove that are relevant to the transfer mechanism (Supplementary Fig. 5, 15). To address the reviewer’s concern we now describe a new set of criteria for the opening between the EC ends of TM3 and TM6. This involves the calculation of distances between center-of-mass of two groups of atoms: (i)- the C_{alpha} atoms of residues 315 to 318 (on TM3), and (ii)-the C_{alpha} atoms of residues 432 to 436 (on TM6). As described in the manuscript and shown in histograms in Fig. 7, lipid translocation between the central and EC regions of the groove requires an opening between TM3 and TM6. As mentioned above, the “*other motions in the protein lumen*” cause a widening are related to

breaking of E313/E318/R432 gates and lipid accumulation at these residues. The new simulations also show (Supplementary Fig. 15c) that lipid flip occurs only under conditions when all the interactions in the network responsible for the gating dynamics (E313-R432, E318-R432, and T333-Y439), are broken. This is described in detail in the revised manuscript in Pages 12-13.

21) Most importantly, there is no experimental data in the paper supporting the role of the salt bridge in gating/closing the pathway. The experiments in which this salt bridge has been manipulated/abrogated all result in a less effective translocation pathway (the opposite effect), rather than providing evidence supporting a more open path. This is a major problem here, and I suggest that this point (gating/breathing motion of the salt bridge) be substantially deemphasized and reduced to perhaps an interesting observation within the simulations, rather than the mainstay of the paper.

As discussed in response to point #17, our experimental data supports the notion that E313, E318 and R432 are critical for scrambling and that their interaction is not static. Further, in the crystal structure this region of the pathway is too narrow to allow lipid penetration consistent with the idea that its dilation (opening) is required for scrambling. Finally, previous MD simulation studies by the Grabe and Tajkhorshid labs suggested that the E313/R432 pair is one of the major hot spots on the protein for lipid recruitment, presumably attracting lipids via electrostatic interactions. Thus, these mutations would impair scrambling as well by decreasing the lipid occupancy of the extracellular vestibule by the lipids we showed are a trigger for the gating mechanism.

Indeed, our simulations of the E313A/E318A/R432A construct (see Figure below) shows that in this triple mutant the extracellular vestibule cannot undergo the lipid-triggered conformational rearrangement we describe for the WT construct. This causes an over-stabilization of the T333-Y439 constriction thus preventing lipid permeation.

While these results are reassuring and reinforce the conclusions regarding the role of the gating mechanism and the relation with all aspects of the experiments, we prefer not to include them in already very dense manuscript since they do not add new mechanistic understanding to the process of nhTMEM16-mediated scrambling.

Figure 1: MD simulations of the E313A/E318A/R432A triple mutant. (Left) Snapshot after 200ns MD simulations of the E313A/E318A/R432A triple mutant nhTMEM16 in POPC membrane. The three mutations were introduced (with VMD mutator plugin) into the protein structure. The mutated residues are shown in blue. T333 and Y439 residues are drawn in purple and green respectively. The lipid that was observed to translocate to the EC side and flip in the wild type protein is shown in licorice. The lipid phosphorous atoms in the EC side of the groove are drawn in gold spheres. (Right) Time-evolution of the distance between Y439 OH and C_β of T333 in the triple mutant trajectory.

22) If the gating mechanism is truly effective, one additional simulation, in which this can be further nailed down (at least within the simulation domain of the study), would be to keep the gate “open” through some restraints and show increased lipid movement (flipping, or at least some diffusive indicators along the pathway or any quantitative measure pointing in the direction of more efficient motion). Such an additional simulation can at least provide evidence on the computational side that the formation of the salt bridge can hamper permeation. Still, in the absence of experimental support for the effect, I would largely reduce the tone on the importance of this feature.

We thank the reviewer for these thoughtful suggestions about additional computations. We hope that new set of simulations and their analyses presented in Responses 17-21 and included in the revised manuscript, offer sufficient convincing evidence.

23) In the A385W simulation, it is stated that the bulky side chain of Trp induces an open conformation of the groove, but in the Q436W simulation, introducing W at Q436 locks the groove in a closed conformation. It is expected that authors provide clarification as to how Trp brings about opposite effects at the two sites in the simulation (perhaps describing/showing the mutation sites in more detail). Unfortunately, both mutants have similar phenotypes in the experiments, namely greatly reducing the scrambling activity (by >100 fold), which would be hard to reconcile with the observed opposite effect of W at these two different sites.

As we described in the original manuscript, in our simulations W385 or W436 form H-bonds with E313, stabilizing non-conductive conformations of the groove (Page 14, Supplementary Fig. 7d, 16a, c). We observe this in 3 out of 4 protomers. This interaction is not present in the wild type protein and adds to the energy cost of opening the lipid translocation pathway in the mutants. In one of the A385W protomers this H-bond is not stable and scrambling occurs. It is worth emphasizing that both mutants do scramble lipids, albeit at greatly reduced rates, and our simulations provide an explanation for the reduced cost: opening of the pathway is more difficult, but when it opens it does so like it does in the WT protein. This is now re-emphasized in the revised manuscript, see Page 14.

24) The authors did extensive W-scanning mutations on residues lining the upper half of the groove; 17 residues have been tested in total. However, not every single residue from the upper groove has been tested. For example, N435 is one of the residues that form the strongest interactions with lipids (Fig. 4). However, this residue is not tested by mutagenesis, while the other weaker interactors were tested. I am fully cognizant of the potential technical difficulties here, and the fact that the authors are already providing a large data set, but considering this mutant would strengthen the connection between the simulations and the experiment.

We thank the reviewer for this suggestion. We introduced the N435W mutant in nhTMEM16 and assessed whether it affects scrambling and/or channel activity. We find that this mutation produces a modest (~10-fold) reduction in the scrambling rate constants in the presence of

Ca²⁺ and has WT-like properties in its absence. Similarly, its channel flux activity is not severely affected. This data is included in the revised Fig. 2 and 3. We find these results consistent with the idea that since the N345 side chain points towards the membrane rather than into the groove (as seen in the X-ray structure and in our MD simulations), it is expected that its mutation would have a small effect reflecting limited importance. Notably, the rotation of TM6 we observe in our simulations moves this side chain even further away from the groove, likely lessening further the impact on lipid and ion transport inferred from the structure alone.

25) Similarly, all the 17 mutated residues are from the extracellular half of the groove even though the authors mention that certain residues from the cytoplasmic half, for example R505, also form strong interactions with lipids inside the groove. This is somewhat surprising. If the authors would like to remain focused only on the extracellular half, in which case it may be helpful to provide a heads up to the reader, so they are not surprised by the lack of any data for the cytoplasmic half.

We thank the reviewer for pointing this out. We only focus on the extracellular side of the groove as others have extensively mutagenized the intracellular side (Yu et al., Elife 2015; Gyobu et al., PNAS 2017). We added a statement to clarify this point in Page 5.

26) Previous studies suggest that phospholipid scrambling does not require any ionic electrochemical driving force, however, the nonselective ion transport by scramblases relies on scrambling and a leak current is associated with lipid transport. In this manuscript, authors found that the charge reverse mutation R432E retains WT-like channel activity while not conducting lipids. The interpretation of this result should be discussed.

We thank the reviewer for highlighting this interesting finding. However, the focus of the present manuscript is on lipid scrambling and the mechanisms regulating opening of the groove rather than ion transport. Therefore, we would refrain from adding mechanistic speculations that are, at present, unfounded in experiments.

27) Based on Fig. 2g-2h, and Fig. 3b, it is stated that the mutations that affect ion channel activity mostly overlap with those that influence lipid scrambling, with the sole exception of N317. The bar plots in Fig. 2g-2h, and Fig. 3b are too compressed and difficult to read. Please separate the bars between each mutant. It's hard to tell any correlation between Fig. 2g-2h and Fig. 3b merely by looking at the figures (for example, S382W severely affects scrambling while does not affect ion channel activity at a similar extent). Analysis of correlation would be helpful to interpret the data. N317W strongly reduces AN317W(0 Ca²⁺) by ~7-fold, while has not much influence on AN317W(+Ca²⁺). Is there a way to reconcile the discrepancy? If not, perhaps just mention it explicitly. I understand that there is a large data set here (and that is great) and not everything can be perfectly fitting, but if there are clear deviations the reader should be alerted.

The reviewer raises an excellent point. However, the different nature of the two assays employed (the scrambling assay informs on the kinetic properties of lipid transport while the flux assay is an end-point measurement of ion channel activity) prevents us from proposing a direct correlation between the two. Furthermore, as we discussed above (point #18 and Page 7 of the revised manuscript) the flux measurement underestimates the effect of the mutations. Therefore, it is not possible to draw a direct correlation between the results of the two assays. To facilitate the comparison between the flux data and the scrambling measurements we have now color-coded the mutated residues in Fig. 3 based on their effect on scrambling (reported in Fig. 2). We discussed this in greater detail on page 8 where we state

“Overall, the mutations that affect ion channel activity mostly overlap with those that influence lipid scrambling: the 4 residues with ‘low’ impact on scrambling have WT-like channel activity in +Ca²⁺, the 4 with ‘medium’ scrambling impact have little effect channel activity. Of the ten loci with high impact on scrambling, mutations of two (A385 and Q436) also significantly reduce channel activity, while five (T381, S382, R432, Y439 and F440) have little to no effect on ion transport. Mutation of the remaining 3 residues has only low to intermediate effects on ion flux, likely because the flux assay underestimates the effect of the mutations”

28) The salt bridge rearrangement involves the breaking of R432-E313 and the bonding of R432-E318. In Suppl. Fig. 7d upper panel, the red trace almost overlaps with the black one, does that mean the R432 is forming bond with E313 and E318 at the same time? What’s the behavior of the lipids in this situation?

Indeed, there are conformations sampled in our simulations (in the original ones as well as in the new simulations added to the revision) where R432 is simultaneously interacting with E313 and E318. This mode of interaction results in a closed EC vestibule, relatively low lipid count of the EC vestibule (all shown in the original manuscript) and does not favor lipid transfer.

29) Could the dilation of the central constriction in the A385W simulation be mainly due to the rotation of Tyr439 (Fig. 6g), rather than the movement of TM3 and TM6. One has to better quantify the separation of the two helices in order to associate the dilation to helix separation. From SFig. 7, T333-Y439 separation happens in several systems. Do they see similar sequential events of R432 sidechain rearrangement and Y439 rotation as described in the A385W simulation?

In our new set of simulations (please see the new Fig. 7, Supplementary Fig. 15 and pages 11-13 of the revised main text) we observed the following dynamic mechanism in multiple trajectories: rotation of Y439 opens the pathway for the lipid to advance from the central part of the groove to the EC side and concomitantly flip. However, in order for the lipid to be released from the protein, i.e. to merge with the EC leaflet, the EC ends of TM3 and TM6 must move apart. This is quantified in the revised manuscript in Fig. 7c, the right-most histogram column.

Minor points

It is stated that the E313-R432 salt bridge in the crystal structure serves as a gate. If I am not mistaken, E313 and R432 do not form ionic bond in the two available nhTMEM16 structures 4WIT.pdb and 4WIS.pdb (authors use the latter for their simulations).

As discussed above (point 2), we modified language throughout the manuscript to remove reference to “salt-bridge” and replace it with either “electrostatic interaction” or simply “interaction”.

Lines 151-153: “...TM4 residues (N317, F330, T333 and V337) have low to no impact on scrambling...” N317 is actually on TM3 rather than TM4. A recent study on nhTMEM16 shows that T333 is functionally important and that mutating this Thr to the corresponding amino acid Val in the TMEM16A channel homolog results in very little scrambling. A brief mention/discussion would be in order.

We apologize for the improper assignment of N317 to TM4, this is now corrected (page 6). We are not sure about the origin of the different role ascribed to T333 in the two studies, it is possible that the hydrophobic Valine side chain disrupts interactions that a Tryptophan side chain does not. This is now discussed in page 7.

Line 288-290: Fig. 6i should be Fig. 6h. What distance is measured between Y439 and lipid? The distance between Y439 and the lipid decreases and then increases as the lipid bypasses Y439, but there is no data showing that Y439 coordinates the headgroup during the lipid translocation. As shown in Fig. 6g, Y439 side chain rotates away toward the membrane rather than coordinating the lipid. Please clarify.

Thanks for this comment. We fixed the reference to Fig. 6h. We measure minimum distance between Y439 and the translocated lipid. This distance, as shown in Fig. 6h goes down to ~2Å, i.e. within direct interaction range. When Y439 side-chain rotates away towards the membrane it no longer coordinates the lipid (the distance increases in Fig. 6h). Thus, there is a time interval (starting at point C in Fig. 6h) when lipid resides at the central region of the groove coordinated by Y439 before the lipid gets released. That Y439 side chain directly coordinates lipid before the lipid is translocated can also be seen in our new simulations added in the revision. This point is now clarified in the revised manuscript, page 11, with the following sentence: “*The process brings to light the essential role of the Y439 side chain that coordinates the headgroup of the flipped lipid throughout the translocation event until the lipid is released to the IC leaflet*”.

What is the average number of lipids in each groove of the individual construct? From SFig. 4a and 4e, A385W has more lipid occupancy than WT1 in PROA, and Q436W has more lipid occupancy than both WT1 and WT2 in PROB. Please explain why these mutants have even better lipid occupancy compared to the WT constructs.

Supplementary Fig 4a and Supplementary Fig 4e show lipid counts in the entire groove of PROA and PROB. These are not the best measures to discriminate our systems from each other, because the numbers do not correlate with any mechanistically important details; without the opening of the various constrictions, lipids do not scramble even if headgroups are present in the groove. These counts are given for comparative reference to the other, mechanistically important loci. As described throughout the original manuscript, and reiterated here, the two systems (from the originally reported simulations) that show accumulation of lipid in the EC vestibule are WT2 and one of the subunits of the A385W construct. As stated, accumulation of lipids is related to breaking of the EC gates as detailed in Fig. 5 of the original manuscript. This relationship between the broken gates, increased lipid numbers and the translocation process has been observed as well in the new set of simulations added to the revised manuscript.

Fig. 7: The R432L mutation is not mentioned in the text.

Thank you for noticing this. We now mention R432L in page 15.

Lines 68-71: Figure 1 only shows TM3-6, TM7 is missing. Isn't it better to also show TM7 along with the other 4 helices in Panel B of Figure 1? It is introduced as one of the cavity-lining helices.

We now show TM7 in Fig. 1b

Which pairs of amino acid were chosen to approximate the width of the groove and why?

The pairs of amino acids and rationale for the choices are discussed in the legend to Figure 1: “Dashed gray lines indicate the distance between the side chains at the extracellular entry (E313/R432), the mid-point and (T333/Y439) at the intracellular vestibule (A356/F463) of the groove.” These residues were chosen as A356/F463 is the widest point of the intracellular

vestibule, while the T333/Y439 and E313/R432 delimit the narrowest portion of the groove indicating that the groove is wide at the intracellular entryway and narrows towards the extracellular portion.

Fig. 2b: No fluorescently labeled (green dots) or bleached (black dots) lipids are shown on the vesicles. Difficult to compare with Fig. 2a.

The numbering of the panels in SFig. 4 is not matching with the description in the text.

In SFig. 8, the upper panel is PROB and lower panel is PROA, while in SFig. 7, the panels are not labeled. I assume the top panels are PROA. Please label the panels and make them consistent. The scaling should also be consistent in SFig. 7.

Lines 255-257: "... (Fig. 5, blue circle denoted by "")..... a rare mode of lipid penetration ... was observed." There is no blue circle denoted by * in Fig. 5. In which monomer of A385W does the rare mode take place? And the time frame for this rare mode?*

The numbering of the panels in SFig. 7 is not matching the description in the text.

There is a mention of Fig MD1 in the text. Not clear what it refers to.

Figure 1. Description of the dashed (solid) lines in panel B is confusing. Fix.

We thank the reviewer for noticing these inconsistencies and mis-labelings in the figures. We have corrected them in the revised manuscript.

REVIEWERS' COMMENTS:

Reviewer #1 (Remarks to the Author):

The authors have addressed the comments of the reviewers in a satisfying manner. The revised manuscript has improved and is now acceptable for publication.

Reviewer #3 (Remarks to the Author):

The revised manuscript has significantly expanded the data set, providing now an even larger amount of valuable information. At the same time, several of the critical points have been addressed, and the quality of the manuscript has been improved. I note, however, that the authors have not addressed a number of major concerns raised during the initial evaluation of the paper, mostly related to the presentation of the results and organization of the manuscript and their story, in which, to my opinion, the connection between the experimental and simulation results is being overstated. Nevertheless, I think there is a ton of very useful information in the study definitely warranting its dissemination to the field. I am happy to recommend the manuscript for publication in Nat. Comm. in the revised form. This will further advance the field of lipid scrambling.